

# What determines the predictability of a Mediterranean cyclone?

Benjamin Doiteau[1,2], Florian Pantillon[1], Matthieu Plu[2], Laurent Descamps[2], and Thomas Rieutord[2,3]

[1]Laboratoire d'Aérologie, Université de Toulouse, CNRS, UPS, IRD, Toulouse, France
[2]CNRM, Université de Toulouse, Météo-France, CNRS, Toulouse, France
[3]Met Éireann, Dublin, Ireland

**Correspondence:** Benjamin Doiteau (benjamin.doiteau@aero.obs-mip.fr)

**Abstract.** Mediterranean cyclones are essential components of the climate in a densely populated area, providing beneficial rainfall for both the environment and human activities. The most intense of them also lead to natural disasters because of their strong winds and heavy precipitation. Identifying error sources in the predictability of Mediterranean cyclones is therefore essential to better anticipate and prevent their impact. The aim of this work is to characterise the cyclone predictability in this region. Here, it is investigated in a systematic framework using European Centre for Medium range Weather Forecasting (ECMWF) fifth generation reanalysis (ERA5) and ensemble reforecasts in a homogeneous configuration over 20 years (2001-2021). First, a reference data set of 2853 cyclones is obtained for the period by applying a tracking algorithm to the ERA5 reanalysis. Then the predictability is systematically evaluated in the ensemble reforecasts. It is quantified using a new probabilistic score based on the error distribution of cyclone location and intensity (mean sea level pressure). The score is firstly computed for the complete data set and then for different categories of cyclones based on their intensity, deepening rate, velocity and on the geographic area and the season in which they occur. When crossing the location and intensity errors with the different categories, the conditions leading to a poorer or better predictability are discriminated. The velocity of cyclones appears to be determinant in the predictability of the location, the slower the cyclone the better the forecast location. Particularly, the position of stationary lows located in the Gulf of Genoa is remarkably well predicted. The intensity of deep and rapid-intensification cyclones, occurring mostly during winter, is for its part particularly poorly predicted. This study provides the first systematic evaluation of the cyclone predictability in the Mediterranean and opens the way to identify the key processes leading to forecast errors in the region.

## 1 Introduction

Extratropical cyclones are fundamental components of weather patterns in the mid-latitudes. The associated frontal systems provide the majority of the beneficial rainfall (Hawcroft et al., 2012) but can also be at the origin of damaging storms (*e.g.* Roberts et al., 2014). A good representation of extratropical cyclones in numerical weather prediction systems is therefore essential to prevent their impacts, and identifying sources of forecast error is an important step to understand the processes leading to a poor predictability and improve forecasts.

In the Mediterranean, cyclones are generally smaller and with shorter life cycle than in other larger basins (Campins et al., 2011). However, they are at the origin of most of the high-impact weather events in the area, including intense rainfall (*e.g.*



Flaounas et al., 2018), windstorms (*e.g.* Lfarh et al., 2023) and compound events (*e.g.* Raveh-Rubin and Wernli, 2016). The location of the Mediterranean between the tropics and the mid-latitudes, as well as the high mountain chains enclosing the basin, make it the site of complex interactions. The influence of Alpine lee-cyclogenesis (Trigo et al., 2002) and Rossby wave breaking coming from the Atlantic (Raveh-Rubin and Flaounas, 2017) is clearly established in the formation of cyclones in the western part of the basin. Mediterranean cyclogenesis can also be influenced by other mountain ranges, the presence of both polar and sub-tropical jets or by the entrance of Atlantic cyclones into the basin (see Flaounas et al., 2022, for a review).

Using a piecewise inversion of the potential vorticity equation, Flaounas et al. (2021) showed that intense Mediterranean cyclones are influenced by two kinds of processes. On the one hand, the intrusion of a potential vorticity streamer in the upper-troposphere, related to deviation of the polar jet and to Rossby wave breaking, is identified as a principal dynamical contribution to cyclogenesis. On the other hand, diabatic processes, and in particular latent heat release, are important in the lower troposphere, where they act as a source of potential vorticity, reinforcing the cyclonic circulation. In some cases, the relatively warm Mediterranean Sea can lead to the formation of tropical-like cyclones, called medicanes, which received interest of the scientific community in the recent years (*e.g.* Miglietta et al., 2021). These phenomena, often poorly predicted, are rather rare with 1 to 2 events per year but can lead to severe rainfall as in the cases of Ianos in September 2020 (Lagouvardos et al., 2022) or Daniel in September 2023.

Limitations in the representation of cyclogenesis processes in numerical weather prediction systems can lead to forecast errors propagating through lead times. Additionally and beyond errors associated with the quality of the numerical model, the chaotic nature of the atmosphere leads to an intrinsic limit of predictability (Lorenz, 1969). More precisely, slight differences in the initial conditions can lead to radically different states of the atmosphere at increasing lead times. The forecast error is therefore due to both limitations in the representation of physical processes in the numerical model and to the chaotic nature of the atmosphere (practical and intrinsic preditcability, respectively; see Melhauser and Zhang, 2012). Extending earlier work by Zhang et al. (2007), Baumgart et al. (2019) identified three phases in forecast error growth. In a first phase, errors in the representation of diabatic processes dominate in the first 12 h lead time. In a second phase, they are projected on the upper troposphere between 12 h and 2 days by tropospheric divergence. In a third phase, after 2 days lead time, the error growth is dominated by the upper-troposphere dynamics.

Ensemble prediction systems have been developed to provide an estimation of the forecast error growth. They offer a measure of forecast uncertainty and different possible scenarios from perturbed initial conditions and model parametrisations (Leutbecher and Palmer, 2008). This is crucial for extreme weather events, which are hardly sampled especially at longer lead times, ensemble prediction providing robuster results than a single deterministic forecast. For these reasons, ensemble prediction systems have long been proved useful for the early detection of extratropical cyclones and their associated hazards (Buizza and Hollingsworth, 2002) or to assess the sensitivity of hurricane genesis to the initial conditions (Torn and Cook, 2013). In the Mediterranean, studies based on ensemble forecasts revealed large uncertainty in the formation of medicane case studies and pointed its origin in error growth along the Rossby wave guide over the North Atlantic a few days ahead (Pantillon et al., 2013; Portmann et al., 2020).





To the best of the authors' knowledge, there is currently no systematic identification of error sources in the predictability of Mediterranean cyclones. For instance, earlier work highlighted the crucial representation of upper-level dynamical precursors in the western Mediterranean (Argence et al., 2008; Vich et al., 2011) or cloud processes and air-sea interactions for medicanes (Miglietta et al., 2015; Tous et al., 2013) but these results relied on case studies. Using ensemble forecasts, Di Muzio et al. (2019) suggested the existence of a predictability barrier for the formation of several medicanes but these rare events may not be representative of the broad spectrum of Mediterranean cyclones. Noteworthy, Picornell et al. (2011) assessed the deterministic forecast quality for more than 1000 cyclones during a whole year and found that the mean error in location increased from 50 km at 12 h to 118 km at 48 h lead time. However, the results were limited to relatively short forecast ranges and were not linked with the cyclone characteristics.

On a broader scale, Froude et al. (2007a, b) were among the firsts to investigate the predictability of extratropical cyclones in a systematical framework. They tracked the cyclones as objects in global forecast data for two winter and two summer periods and defined errors in both location and intensity based on maximum vorticity compared to analysis data. For the location, they found out that the error increases almost linearly at a rate of 1.25 geodesic degree per day. For the intensity, they highlighted differences between summer and winter cyclones. In particular, intense storms occurring during the winter period were more poorly predicted and this was attributed to an incorrect representation of their vertical structure. More recent studies followed a similar approach and showed a systematic slow bias in the position of North Atlantic cyclones and a weak bias in intensity of the deepest ones (Pirret et al., 2017; Pantillon et al., 2017). They explored links between the predictability and the dynamics of cyclogenesis but faced a robustness issue due to limited samples.

In this paper, ensemble reforecasts are used to systematically identify errors in the location and intensity of Mediterranean cyclones. The forecast model covers a 20-year period with a homogeneous configuration, which allows to extract robust statistical signals. The aim of the paper is to characterise the cyclone predictability in the Mediterranean region. Their representation in the ensemble prediction system is discussed and the cyclone characteristics leading to a poorer or better predictability are identified. In particular, errors in the prediction of the cyclone location and intensity are evaluated for several categories of cyclones, based on their geographical location and seasonality, their intensity, deepening rate and velocity.

The article is structured as follows. In Section 2, the data, tracking methods and tools to evaluate the predictability are described. The catalogue of Mediterranean cyclones and the associated climatology is presented in Section 3. The predictability is then firstly evaluated for the whole data set in Section 4, and secondly for specific categories of cyclones in Section 5. Section 6 contains a summary of the main results and a conclusion of the study.

## 2 Data and methods

### 2.1 Data for the reference tracks: ERA5 reanalysis

Reanalyses assimilate historical observation data spanning decades with a fixed assimilation scheme and forecast model. ERA5 (Hersbach et al., 2020) is the fifth reanalysis produced by the European Center for Medium Range Forecasts (hereafter ECMWF). It is based on the Integrated Forecast System (IFS, cycle 41r2), and includes models for atmosphere, land





surface and ocean waves. The horizontal resolution of the atmospheric model is about 31 km at mid-latitude and it has 137
vertical levels from the surface to 0.01 hPa. The reanalysis products are available globally with hourly resolution, from 1940 to
95 present. In this study, ERA5 is used from 2001 to 2021 with 0.25° horizontal resolution to produce a reference set of cyclone
tracks on a domain covering the Mediterranean (25° N - 50° N, 15° W - 45° E; see Fig. 1).

## 2.2 Tracking method for the reference tracks: the Ayrault algorithm

Before investigating the predictability of Mediterranean cyclones, the first need is to produce a reference set of tracks. The
tracking method is based on the Ayrault (1998) algorithm, which has been implemented in the open-source TRAJECT software
(Plu and Joly, 2023). Originally designed for Atlantic cyclones in coarser model data, the Ayrault (1998) algorithm required
a specific tuning for this study. Indeed, Mediterranean cyclones are generally smaller and with shorter life cycles than in the
Atlantic (Campins et al., 2011) and ERA5 has a higher spatiotemporal resolution than any previous reanalysis used with the
algorithm. Therefore, the parameters have been tuned specifically for both ERA5 and the Mediterranean region.

The main idea of Ayrault (1998) is to track cyclones in the relative vorticity field at 850 hPa first, and then operate a pairing
with the mean sea level pressure (MSLP) field. In the following, a time step is denoted by $t$, the relative vorticity field at
850 hPa by $\zeta$, and the zonal and meridional wind fields by $u$ and $v$, respectively. The Ayrault (1998) algorithm can be separated
in five steps:

(1) Data preparation: a moving average with Gaussian weights is applied to $\zeta$ and to $u, v$ at 850 hPa and 700 hPa to remove
noisy features in these fields. The characteristic length in the weight decay is 225 km for $\zeta$ and 280 km for the wind
fields.

(2) Detection of $\zeta$ maxima: local maxima are detected in the $\zeta$ smoothed field and a single maximum (the strongest) is
retained within a radius of 300 km.

(3) Loop on successive time steps: for every $\zeta$ maximum at time $t$, a corresponding maximum at time $t + 1$ is searched
for using a three steps method. First, the $\zeta$ maximum at time $t$ is advected by the wind at both 850 hPa and 700 hPa,
giving two guess positions. In a second step, a new $\zeta$ maximum at time $t + 1$ is searched for in the neighbourhood of the
two guessed points. Third and last, a quality criterion selects the best new $\zeta$ maximum by taking into account both the
distance between the guessed point and the new $\zeta$ location and the $\zeta$ value variation. A cyclone track is finally defined
by the successive positions of $\zeta$ maxima at every time step.

(4) Pairing with MSLP: for every point belonging to the track, if a local minimum of MSLP is located in a 3° square centered
on the $\zeta$ maximum, it becomes the new track point. The $\zeta$ maximum remains the track point in the opposite case.

(5) Validation criteria: the tracking process is stopped when the value of the $\zeta$ maximum is less than $10^{-4} s^{-1}$ or of the
MSLP minimum is greater than 1015 hPa. Among all tracks, only those which last for longer than 24 h and reach at least
1005 hPa along their life cycle are retained. This last criterion avoids most of the artefact cyclones. Indeed, most of the
cases with deepest MSLP over 1005 hPa appear to be secondary local lows caused by strong storms crossing northern




Europe. Finally, an additional criterion is applied to only retain tracks that enter into the Mediterranean or the Black Sea areas.

The Mediterranean-adapted version of the Ayrault (1998) algorithm has been successfully tested with a slight different configuration in an intercomparison of 10 tracking methods using ERA5 (Flaounas et al., 2023). The produced data set remained close to the consensus between all algorithms in the spatial and seasonal distribution of cyclones. In the present study, our data
set is used as a reference instead of the consensus produced by Flaounas et al. (2023) for two principal reasons. First, the latter contains only 206 tracks in the highest confidence level, which is not enough for a systematic study. Second, Ayrault (1998) is conceptually similar to the tracking algorithm applied to the reforecasts (see Section 2.4.), which reduces the influence of the tracking method on the results to focus on the predictability.

### 2.3    Data for the predictability study: IFS reforecasts

Reforecasts are forecasts made retrospectively on a historical period spanning typically several decades with a fixed model version. While these properties are shared with reanalyses, reforecasts are different in that they do not assimilate any observation beyond initial conditions, making them comparable to operational forecasts, except that they are in the past. They are therefore a key tool to investigate the predictability of the Mediterranean cyclones previously tracked in ERA5. The ECMWF ensemble reforecasts used here are constituted of 10 perturbed + 1 control members based on the IFS model (cycle 47r3) and initialized
from ERA5 (Vitart et al., 2019). Initial perturbations on the reanalysis are constructed from the ERA5 ensemble data assimilation and singular vectors. Additionally, the model uncertainties are represented using a stochastically perturbed parametrisation tendency scheme (Buizza et al., 1999). The reforecasts used here cover an historical period of 20 years from October 2001 to October 2021, during which they are initialised every Monday and Thursday at 0000 UTC, leading to a total of about 2000 base times. The output spatial resolution of 0.25° is identical to the one in the ERA5 reanalysis. For each base time, a forecast
output is available every 6 h (temporal resolution coarser than ERA5). Despite the maximum lead time of 14 days available with constant resolution in the reforecasts, the maximum lead time is restricted in this study to 144 h because of the short life cycle of Mediterranean cyclones, considering that only 0.07% of the cyclones of our reference data set last for longer than 6 days. The small number of ensemble members able to produce cyclone tracks at longer lead times (see Section 4.1) is also pleading in favour of a limitation of the maximum lead time. Note that the same cyclone can be tracked in several successive
base times. More specifically, this is the case of cyclones with lifetimes longer than 144 h or starting at late lead times and persisting beyond the next base time. When this happens, the forecast tracks are treated independently.

### 2.4    Tracking method for the predictability study: the VDG algorithm

In the reforecasts, the tracking of the cyclones is made with another algorithm (van der Grijn (2002); hereafter VDG) developed at the ECMWF and originally designed for the operational tracking of tropical cyclones. The VDG algorithm, also implemented
in the open-source TRAJECT software (Plu and Joly, 2023), is similar to the previously applied Ayrault (1998), as it also uses MSLP, the $\zeta$ smoothed field at 850 hPa and the horizontal wind at 850 hPa and 700 hPa. The main difference between the




two algorithms is that VDG starts the tracking from a given geographical point or from an existing track. This characteristic is particularly useful when it comes to track predicted cyclones that were previously identified in the reference data set. Applying Ayrault (1998) to the reforecasts would have required an additional step for matching the cyclones, bringing more complexity and subjectivity.

At initialisation, a $\zeta$ maximum is searched for in the reforecast field in the neighbourhood of the reference track calculated in ERA5. The tracking method in VDG is then based on a combination of past movement and steering flow vector $V_{av}$ determined by a combination of the local wind fields at 850 hPa and 700 hPa. In the following, $r$ and $r_{fg}$ are respectively the position of the cyclone and of the first guess. The initial step apart, the VDG algorithm can be divided as follows:

(1) First guest: the steering flow $V_{av}$ and the past movement $r(t) - r(t-1)$ vectors are combined to obtain the first guess position of the next tracking point $r_{fg}$ using the equation $r_{fg}(t+1) = r(t) + w[r(t) - r(t-1)] + (1-w)V_{av}\delta t$, where $w$ is a weight parameter ranging from 0 to 1 depending on the temporal resolution and here set at 0.5, and $\delta t$ is the time step. *N.B.* at the first time step only the steering flow vector is used (there is no past movement).

(2) Detection of $\zeta$ maxima: a maximum is searched for in the $\zeta$ field within a square of 3.5° around the first guess.

(3) Pairing with MSLP: another search is performed for the MSLP minimum using a square of 3.5° centered around the $\zeta$ maximum. The location of this MSLP point finally becomes the next track point $r(t+1)$.

(4) Stopping criteria: the tracking of the cyclone is stopped when the value of the vorticity maximum $\zeta$ is less than the corresponding threshold of $10^{-4}s^{-1}$ or when the value of the MSLP minimum is greater than 1015 hPa, as in the Ayrault (1998) algorithm. This last criterion also implies that the tracking begins only if a MSLP minimum is found below the pressure threshold.

Cyclones detected in the reanalysis are linked with the reforecast by construction of the VDG algorithm, as the position of the cyclone in the reforecast at the initial time, $r(0)$, is directly dependent on the presence of a reference track at the same time.

## 2.5 Predictability metrics

In this study, predictability is investigated using error and spread in both location and intensity. The relationship between error and spread is first used to verify the ensemble calibration before proceeding with further quantification of the predictability. For a well-calibrated ensemble, one should expect error and spread to be comparable in magnitude with each other, whether in location or in intensity.

In the following, errors are defined for each cyclone track by comparing the location or the intensity of ensemble members with the corresponding reference track in ERA5, at each time $t$ of the cyclone life cycle. The spread is for its part calculated from the pairwise difference between the members of the ensemble.

To assess the predictability of the cyclone location, we use the total track error (TTE) as defined in (Froude et al., 2007b; Leonardo and Colle, 2017). The TTE is decomposed into an along-track error (ATE) and a cross-track error (CTE). A positive (respectively negative) ATE stands for a forecast track ahead (respectively behind) of the reference track, while a positive



(respectively negative) CTE stands for a forecast track on the left hand side (respectively on the right hand side) of the reference
track. Track errors (TTE, ATE and CTE) are defined for each individual member and are presented in Section 4. Additionally,
an $\overline{TTE}$ is here defined for each cyclone as the mean of the TTEs over the members at each time $t$ of the cyclone life cycle.
The spread in location (hereafter $\sigma_{loc}$) is for its part determined by averaging the distance between each pair of members as
follows:

$$\sigma_{loc}(t) = \frac{1}{N(N-1)/2} \sum_{1 \leq i < j \leq N} d(r^i(t), r^j(t)) \tag{1}$$

where $N$ is the number of members in which the cyclone is detected by the tracking algorithm at time $t$, $r^i$ (respectively $r^j$)
is the position of the cyclone in the $i$-th member (respectively in the $j$-th member) and $d$ is the geodesic distance between the
two positions.

Regarding cyclone intensity, MSLP errors (hereafter MSLPE) are defined as the difference between the MSLP of each
member and the MSLP of the reference track. Unlike errors on the location, errors on MSLP can also be negative. Consequently,
⟨MSLPE⟩ is defined as the root mean square of the MSLPEs over the members, for a specific event at a specific time $t$ of the
cyclone life cycle. The spread in MSLP (hereafter $\sigma_{int}$) is for its part determined from the root mean square of the differences
between each pair of members as follows:

$$\sigma_{int}(t) = \sqrt{\frac{1}{N(N-1)/2} \sum_{1 \leq i < j \leq N} (p^i(t) - p^j(t))^2} \tag{2}$$

where $p^i$ (resp. $p^j$) is the MSLPE of the $i$-th member (resp. $j$-th member).

An additional metric is defined to compare distributions of TTE or MSLPE between different categories of cyclones (see
Section 5). In a preliminary step, for each category of cyclone, a cumulative density function (CDF) of errors is constructed
by taking into account every member of every cyclone track found at each lead time $\tau$. CDFs of errors are then compared in
a framework close to the continuous ranked probability score (CRPS) described in Candille et al. (2007). The metric denoted
here by CDFE measures the distance between a CDF of errors and a virtual null-error distribution (100 % of the errors equal
to 0):

$$CDFE(\tau) = \int [F_\tau(x) - 1_{x \geq 0}]^2 \, dx \tag{3}$$

where $F_\tau(x)$ is the CDF of the errors (either TTEs or MSLPEs) at a specific lead time $\tau$. 1 stands for the Heaviside step
function. Note that the CDFE metric has the same dimension as the variable on which it is applied A higher CDFE (respectively
the smaller) indicates a poorer (respectively a better) predictability. At each lead time, the statistical significance is evaluated
using the Kolmogorov-Smirnov test, which in our case determines if two CDF of errors are similar (respectively different) at a
confidence level of 95%. This will ensure the robustness of the difference in the predictability of several category of cyclones
presented with the CDFE metric.





## 3   Climatology of the reference data set

This section provides the climatology of our reference data set, based on the Mediterranean cyclones tracked with the Ayrault
(1998) algorithm in ERA5 data. In particular the spatial distribution, the seasonal cycle, the intensity and the velocity of
cyclones are presented. Figure 1a shows the ground elevation over the Mediterranean and toponyms that will be used in this
manuscript.

### 3.1   Spatial distribution

For the whole 2001-2021 period a total of 2853 cyclones are detected in the Mediterranean region, *i.e.* about 140 cyclones
per year on average. The color shading in Figure 1b accounts for the number of tracks having at least one track point within a
radius of 100 km divided by the total number of tracks. The figure can thus be seen as the spatial distribution of the cyclones
of our reference data set regardless of their stage of development. This spatial distribution is not homogeneous, as the majority
of cyclones are concentrated in preferred regions. In particular, six regions of interests designed to cover equal areas are here
identified by visual examination of the spatial distribution.

The six preferred regions concentrate 63 % of the cyclones of the data set. The most active of them is the West Mediterranean
(21 %). It includes the Gulf of Genoa, in the lee of the Alps, which is recognized as the most cyclogenetic area. Then come
the regions of the East Mediterranean (11 %), the Adriatic (10 %), the Black Sea (8 %), the Sahara (7 %) and the Middle
East (6 %). The importance of the Alps in the formation of the West Mediterranean cyclones is clearly established (Trigo
et al., 2002). Horvath et al. (2008) show that lee cyclogenesis is also the dominant formation process for Adriatic cyclones,
whether they form in the Gulf of Genoa or in the Adriatic itself. The same orographic processes are known to play a role in the
formation of Saharan cyclones in the lee of Atlas mountains (Winstanley, 1972; Alpert and Ziv, 1989), while Thorncroft and
Flocas (1997) and Prezerakos et al. (2006) mostly highlighted the importance of interactions between the polar and the sub-
tropical jets in cyclogenesis in this particular region. For the Black Sea, and generally in the eastern parts of the Mediterranean,
Trigo et al. (2002) argued that cyclones are formed by different processes. In particular, they stated that surface cyclones in the
Black Sea seem to be associated with an upper trough in the west of the region, advecting vorticity toward a relatively warm
sea. Similar processes are found in the Aegean. The same authors argued that cyclones in the Middle East are the manifestation
of the extensions of the Asian trough in late spring.

The overall spatial distribution of our data set is in agreement with previous studies (Alpert et al., 1990; Trigo et al., 1999;
Maheras et al., 2001; Campins et al., 2011; Lionello et al., 2016; Aragão and Porcù, 2022; Flaounas et al., 2023). However, two
minor differences remain. First, the hotspot in the western Atlas mountains and the high density of cyclones over the Iberian
Peninsula described in the literature do not appear here. This is mainly due to the criteria used to construct our data set by
removing weak thermal lows with a pressure threshold at 1005 hPa on the one hand, and removing cyclones that do not enter
over the sea on the other hand. Second, the high density of cyclones found here in the Adriatic is not highlighted in the majority
of previous studies.



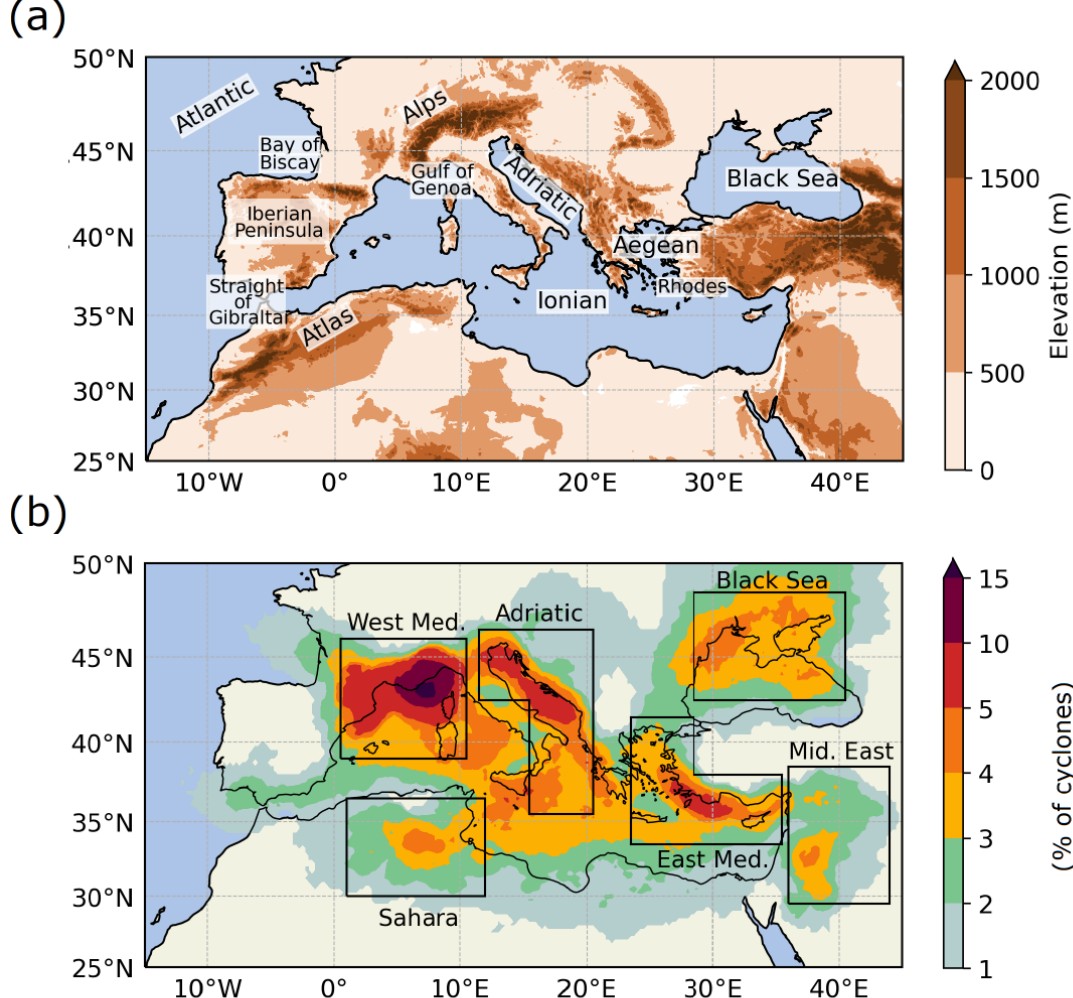

**Figure 1.** (a) Elevation map over the Mediterranean domain with toponyms mentioned in the text. (b) Spatial distribution of Mediterranean cyclones based on ERA5 over the 2001-2021 period, defined as the percentage of cyclones having a track point within a radius of 100 km. Regions of interest are framed by the black boxes.

## 3.2 Seasonal cycle

Figure 2 shows the number of cyclones striking any of the six regions of interest during each month of the year, averaged over the 20 years of our data set. One can see that the number of cyclones in the Mediterranean is highly dependent on the season. The peak activity spans from November to May, while the period from June to October experiences less occurrences. However, this general trend is also dependent on the region considered.





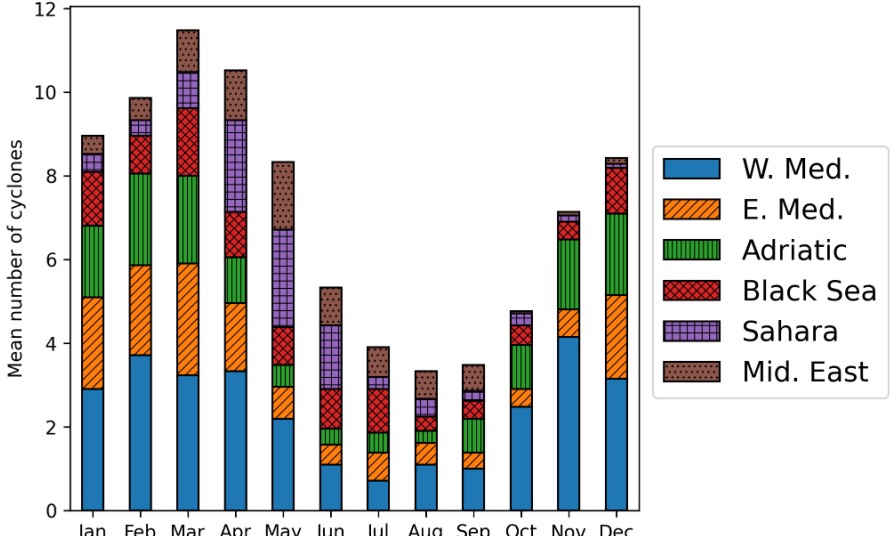

**Figure 2.** Monthly number of cyclones in their mature phase in the six regions defined in Fig. 1b (averaged on the 20-year period).

In the West Mediterranean and in the Adriatic, the cold season generally experiences more cyclones. Horvath et al. (2008) came to the same conclusion for the majority of Adriatic cyclones, while highlighting the importance of a subcategory of summer cyclones for their association with high-impact weather. In the East Mediterranean, more cyclones are also found during the cold part of the year. Saharan cyclones clearly exhibit a peak of occurrence in April and May, in agreement with previous studies (Winstanley, 1972; Alpert et al., 1990; Trigo et al., 2002). The Black Sea has a unique seasonal cycle, with

higher activity during a long period spanning from December to July with a peak in March. The presence of those cyclones during a large part of the year was already observed in Trigo et al. (1999). For the case of Middle East cyclones, higher occurrences are found here from March to June, with a flat peak activity in May, while Trigo et al. (2002) found the peak of activity in August.

### 3.3   Intensity and deepening rate

Figure 3a shows the spatial distribution of the 10% deepest cyclones of the reference data set. They are concentrated in the West Mediterranean, in the Adriatic and in the north-western parts of the Black Sea. The West Mediterranean and the Adriatic are also two hotspots for rapid intensification when looking at the deepening rate (not shown). While cyclones in these two areas are strongly influenced by the Atlantic, the origin of deep cyclones in the western Black Sea remains unclear. In this last region, cyclones do not experience rapid intensification, suggesting other processes of cyclogenesis. The shallowest cyclones

are for their part concentrated in the Gulf of Genoa and in the eastern parts of the Black Sea (not shown). The other shallow cyclones are found mainly in the West Mediterranean, highlighting the wide spectrum of intensities and deepening rates in this particular region.





Figure 3b presents the typical seasonal cycle for three intensity-based categories of Mediterranean cyclones. Shallow and medium-intensity cases are more present during early spring and exhibit a flat minimum from July to November. Deepest cyclones show a much more pronounced seasonal cycle, with very few cyclones during the warm part of the year, and a peak of activity from November to March. Similar characteristics are observed in terms of deepening rate, where slow and medium-intensification categories are more present from December to May and rapid-intensification cyclones are found almost exclusively during the cold part of the year (not shown).

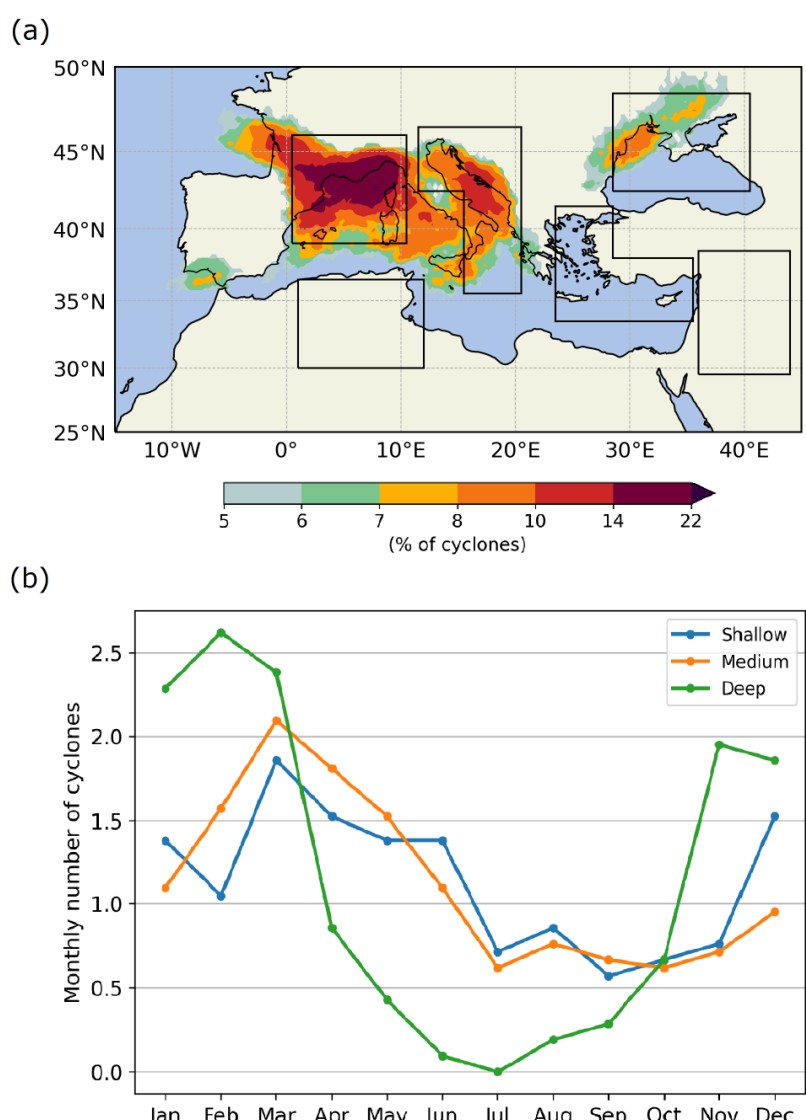

**Figure 3.** (a) Spatial distribution (as defined in Fig. 1b) of the 10 % deepest cyclones of the reference data set. (b) Monthly mean number of cyclones in the 3 categories of intensity. Each category contains 10 % of the tracks data set, i.e., the 10 % deepest (green curve), the 10 % around the median intensity (orange curve) and the 10 % shallowest cyclones (blue curve).

## 3.4 Velocity of Mediterranean cyclones

The velocity of a cyclone is defined here by the median speed along its whole life cycle. According to our calculations on the reference data set, Mediterranean cyclones generally move from the west to the east at a median velocity of 27 km h$^{-1}$.





However, the variability is large and the fastest 5 % are moving at velocities greater than the twice the median. Figure 4 shows the spatial distribution of cyclones in each of the three velocity categories: the 10 % fastest (Fig. 4a), the 10 % around the median speed (Fig. 4b) and the 10 % slowest (Fig. 4c). The strong change in spatial patterns from one velocity-based category

to another highlights the close relationship between the velocity of Mediterranean cyclones and their spatial distribution.

The fastest cyclones (Fig. 4a) can be found in several particular areas. First, cyclones originating from the Sahara are clearly marked along an axis from the south of the Atlas mountains to the Ionian Sea. Cyclones in this region are also the fastest with a median speed of 30 km h$^{-1}$. This result is in agreement with previous studies, which often highlight the high velocities of Saharan cyclones compared to other Mediterranean lows (Alpert and Ziv, 1989; Kouroutzoglou et al., 2011). Second, fast

Atlantic cyclones enter into the West Mediterranean, mainly from the Bay of Biscay and to a lesser extent through the straight of Gibraltar. Third, another group of fast cyclones crosses the western Black Sea. Fourth and last, two other favourable regions for fast cyclones are found in the northern Adriatic and in western Greece. Medium speed cyclones (Fig. 4b) are for their part mainly located over sea, in the West Mediterranean, in the southern Adriatic and in the Ionian Sea. Finally, the slowest cyclones (Fig. 4c) are clearly concentrated in the Gulf of Genoa, with median velocity around 18 km h$^{-1}$. Some quasi-stationary lows can also be found around the island of Rhodes and in the eastern parts of the Black Sea. The location of these last quasi-

stationary lows is contrasting with the fast cyclones observed over the western Black Sea (Fig. 4a), suggesting two different processes of cyclogenesis in the Black Sea region.



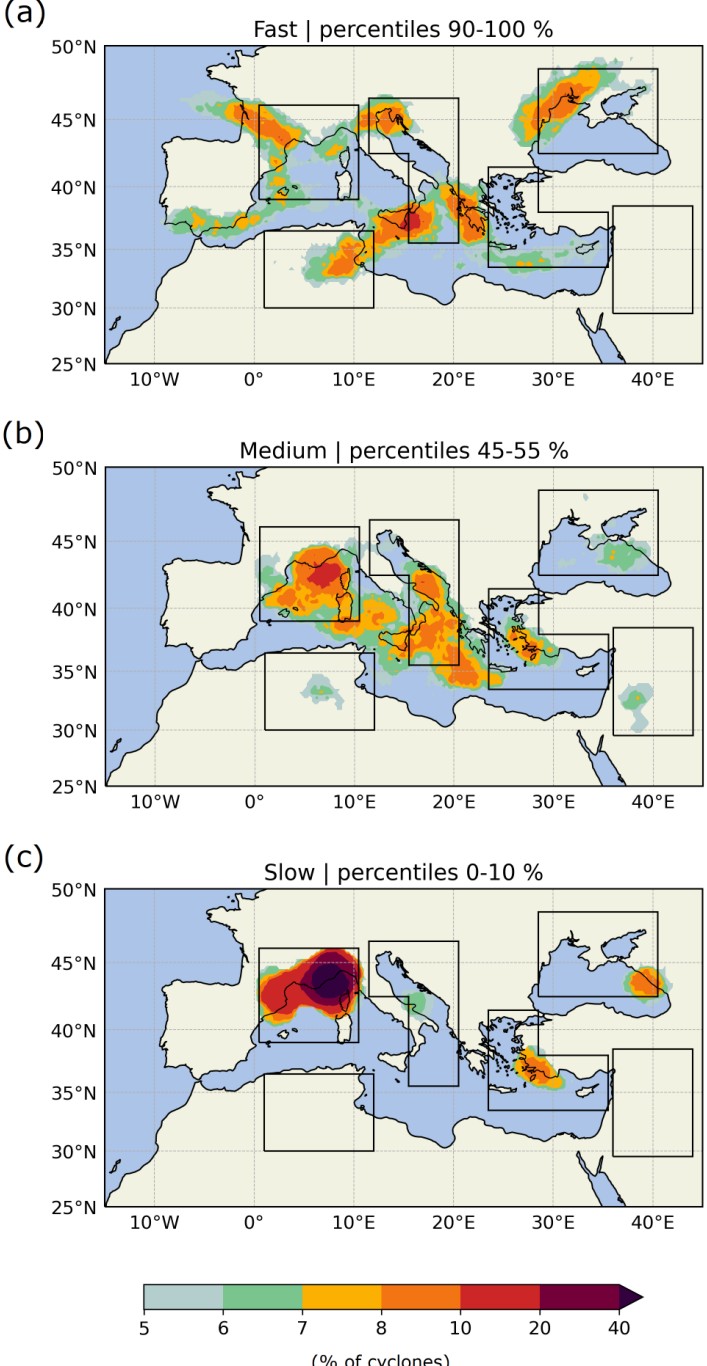

**Figure 4.** Spatial distribution (as defined in Fig. 1b) of three velocity-based categories. Each category contains 10 % of the cyclone tracks data set, i.e., the 10 % fastest (a), the 10 % around the median speed (b) and the 10 % slowest (c). The black boxes are the regions of interests defined in Fig. 1b

.



## 4 Evaluation of the ensemble reforecasts

This section is dedicated to the evaluation of the Mediterranean cyclones representation in the reforecasts. Errors in the location
and intensity, as defined in Section 2, are firstly evaluated by taking the tracks detected in ERA5 as reference, while the
reliability of the ensemble reforecasts is assessed in a second step.

### 4.1 Errors on tracks: location and intensity

To evaluate the reforecasts, both errors in location and in intensity are considered. In Fig. 5, distributions of errors are computed
at each lead time by taking into account the individual error of each member of the ensemble reforecasts, for the all data set.
The very large number of 2853 cyclone tracks ensures these results to be statistically significant. The mean number of members
in which a cyclone is found decreases linearly with lead time (orange curve on Fig. 5). While more than 9 members out of 11
detect a cyclone at the initial time, less than 4 members are remaining on average after 144 h lead time.

The TTEs distribution is presented for each lead time until 144 h (Fig. 5a). Both median error and interquartile range are
increasing with lead time, with 50 % of the TTEs spanning from 80 km to 220 km after 3 day lead time. It is noticeable that the
error growth is not fully linear and seems to exhibit two phases: in the first 78 h, the median TTE growth rate is about 40 km per
day, while from 84 h lead time and beyond it increases at a smaller rate of about 18 km per day. This behavior can be explained
by two different processes. Firstly, the construction of the VDG algorithm constrains the tracking to begin in the neighbourhood
of the reference track. Consequently, the error in location is generally smaller at the beginning of a cyclone track than at its
end. When the lead time increases, the proportion of cyclones followed from early lead times (*i.e* with higher errors) decreases
in comparison of the ones followed from a few hours (*i.e.* with smaller errors). It results that the error growth tends to decline
with increasing lead time. The second process deals with the error saturation. At very long lead times, a forecast is comparable
to a random climatological state of the atmosphere. Consequently, the mean and median errors are expected to increase at a
smaller rate at long lead times and saturate ultimately at a constant value.

Overall, the maximal growth rate of 40 km per day in the first 78 h lead time is remarkably close to the 43 km per day
found by Picornell et al. (2011) in the Mediterranean. The authors used for their part the ECMWF operational deterministic
model during the 2006-2007 period and evaluated errors only during the first 48 h, which may explain the comparable error
growth despite the older model version used in their study. In the whole Northern Hemisphere, and using the operational
ensemble prediction system of the ECMWF from January to July 2005, Froude et al. (2007b) found a much higher mean error
growth rate of 1.25° (about 137 km) per day and almost constant until 7 days lead time. The coarser resolution of the ensemble
prediction system used in their study (about 80 km) or the particular characteristics of Mediterranean cyclones could explain
this difference in the mean error growth rate.

As presented in Section 2, the TTE can be decomposed into ATE and CTE. In our case, the ATE exhibits a weak and constant
bias of -20 km from 60 h lead time and beyond, indicating that forecast tracks are slightly late compared to the reference (not
shown). It is in agreement with Froude et al. (2007a), who highlighted that forecast cyclones are in average getting too slow by
about 1 km per hour compared to the analysis. For their part, Pirret et al. (2017) and Pantillon et al. (2017) found a systematic



slow bias in the prediction of 60 and 25 severe European storms, respectively. The little bias found here in the ATE, however, is much smaller than in the previous mentioned studies. Regarding the CTE a weak positive systematic bias is observed, growing at a constant rate of 5 km per day, indicating a weak shift to the left of the track (not shown). When looking into absolute values of ATE and CTE, it appears that the TTE is the result of an equivalent contribution of both components.

Errors in intensity (MSLPE) are presented in Fig. 5b. A very weak bias of -0.2 hPa per day is observed when looking at the mean, pleading in favour of a well-centered distribution of the ensemble reforecasts around the reference with a slight overestimation of the cyclone intensity, as in Froude et al. (2007b). After 3 days of forecast, 50 % of the MSLPEs are between -2.5 hPa and 1.5 hPa and the interquartile range grows linearly until the last lead time of 144 h. When looking at absolute MSLPE (not shown), a little positive linear bias of 0.5 hPa per day is observed. Froude et al. (2007b) highlighted an even

smaller bias around 0.2 hPa per day for the whole Northern Hemisphere. It could indicates a better prediction of the intensity of cyclones in other basins compared to the Mediterranean, however, the small magnitude of these biases should be considered.

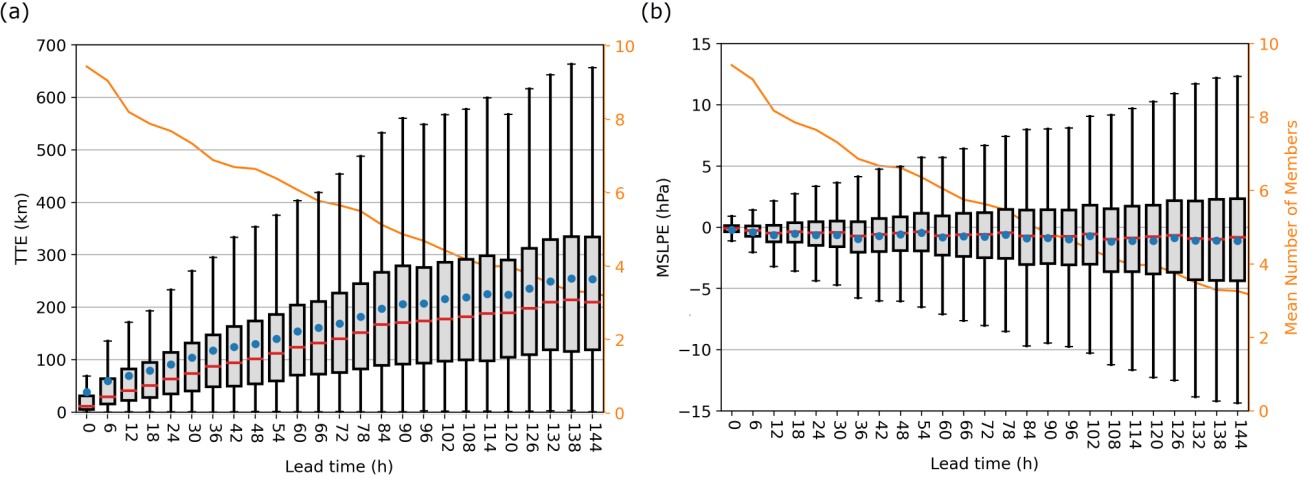

**Figure 5.** Distributions of (a) total track errors (TTEs) and (b) MSLP errors (MSLPEs) compared to ERA5 in function of the lead time. Means are depicted by the blue circles, medians by the red lines, the first and third quartiles by grey boxes, and the minima and maxima by black lines. The orange curve is the mean number of members in which a cyclone is detected.

## 4.2    Reliability of the ensemble reforecasts

The reliability of the ensemble reforecasts is evaluated for both cyclone intensity and location by comparing for each event and at a specific lead time the spread and the mean error of the members, as defined in Section 2. One expects the mean error to be

close to the spread for a well-calibrated ensemble, while a mean error greater (respectively smaller) than the spread indicates an under-dispersive (respectively over-dispersive) ensemble prediction system.

Figure 6 presents a comparison between the spread and the mean error of the ensemble at 72 h lead time, for the location in Fig. 6a and for the intensity in Fig. 6b. Similar observations can be made in both cases: firstly, the ensemble is reasonably





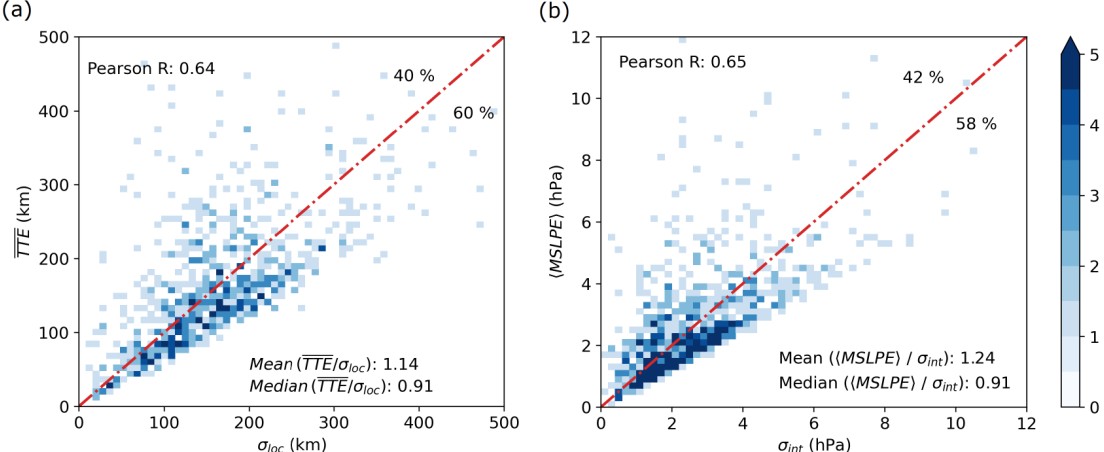

**Figure 6.** For each cyclone at 72 h lead time, the blue shading represents the number of events per bin. (a) Mean of the TTEs of the ensemble members, denoted $\overline{TTE}$, compared to the spread in location, denoted $\sigma_{loc}$. The bin length is equal to 8 km. (b) Root mean square of the MSLPEs of the ensemble members, denoted by $\langle MSLPE \rangle$, compared to the spread in intensity, denoted $\sigma_{int}$. The bin length is equal to 0.2 hPa. The red curve represents an idealised, perfectly-calibrated set of ensemble reforecasts with a mean error equivalent to the spread. Percentages indicate the proportion of events in each half part.

reliable with an identifiable linear trend between spread and mean error (correlation coefficient around 0.65). Secondly, there
is a slight but noticeable over-dispersion with 60 % of events presenting a spread greater than the mean error. Finally, when looking at the mean and median error-over-spread ratios, it appears that the mean is greater than 1 in both cases, equal to 1.14 for the location and 1.24 for the intensity, while the median ratio is equal to 0.91 in both cases. This indicates that while the ensemble is slightly over-dispersive for most of the cyclone cases, some of them remain very poorly predicted with a mean error much greater than the spread. It is noticeable that the opposite case with a spread much greater than the mean error is not
observed. These three conclusions remain valid for the different lead times (not shown).

# 5 Predictability of different types of Mediterranean cyclones

In the previous section the predictability was evaluated considering the complete data set. In this section, cyclones are categorised following different features in order to determine the factors leading to a systematically better or poorer predictability. In particular, differences in the predictability are identified in the spatial distribution, the seasonality, the intensity and the
velocity of cyclones.

For each cyclone categorisation, CDFs of errors in both location and intensity are used to compute the CDFE metric presented in Section 2 at a specific lead time. Note that the CDFE having the same unit as the variable considered, the greater the CDFE (respectively the smaller), the poorer (respectively the better) the predictability of the cyclone category. To illustrate





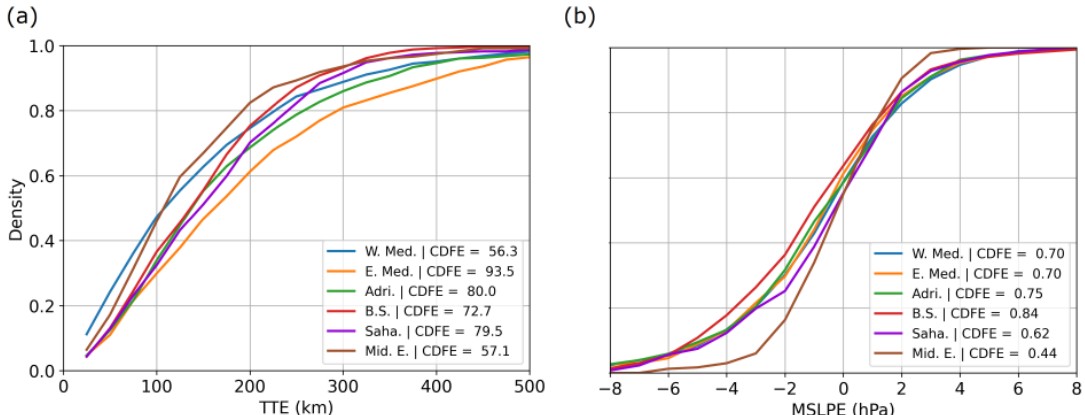

**Figure 7.** CDFs of the errors in location (a) and in the intensity (b) for the six regions defined in Fig. 1b at 72 h lead time.

the approach, Fig. 7 presents CDFs of errors for the six regional categories presented in Section 3. On this representation, a
category of cyclone is better predicted than another when its CDF is closer to the Heaviside step function. At 72 h lead time
and for the TTE (Fig. 7a), the East Mediterranean is the region in which cyclones are the most poorly predicted (orange curve),
while the Middle East and West Mediterranean cyclones have the smallest errors (brown and blue curves). This is highlighted
by the CDFE metric, with scores ranging from 57.1 km for the Middle East and 56.3 km for the West Mediterranean to 93.5 km
for the East Mediterranean. In terms of MSLPE (Fig. 7b), Middle East cyclones are once again the best predicted with a CDFE
of 0.44 hPa, while the Black Sea is the region in which the intensity of cyclones is the most poorly predicted at this particular
lead time, with a CDFE equal to 0.84 hPa. In the next sections, the CDFE score is computed at each lead time in order to
compare the predictability between several categories of cyclones along the complete forecast duration.

## 5.1 Regional differences

As shown in Fig. 1b, the spatial distribution of Mediterranean cyclones is not homogeneous, and six regions have been defined
according to their cyclone density. Figure 8a presents the differences in the predictability of cyclone location using the CDFE
metric applied at each lead time on the TTEs distributions of the six regions (color curves). It immediately appears that cyclones
in the West Mediterranean are the best predicted at lead times beyond 42 h. The statistical significance of differences between
this region against any other is verified, except with the Middle East at 66 -90 h and 126 -138 h lead time. Another interesting
feature is the apparent diurnal cycle for Saharan and Middle East cyclones (purple and brown curves) between 72 h and 120 h
lead time. Peaks of errors are visible at 84 h and 108 h for the Sahara and 90 h, 108 h and 132 h for the Middle East. Forecasts
being always initialised at 00 UTC, these peaks of errors in the location of cyclones are happening during the warm part of the
day for the both regions. Finally, the best and worst categories are shown for illustration on Fig. 8c. The apparently poorest
predicted categories, namely the Adriatic and East Mediterranean cyclones, are in fact following the general behavior of the




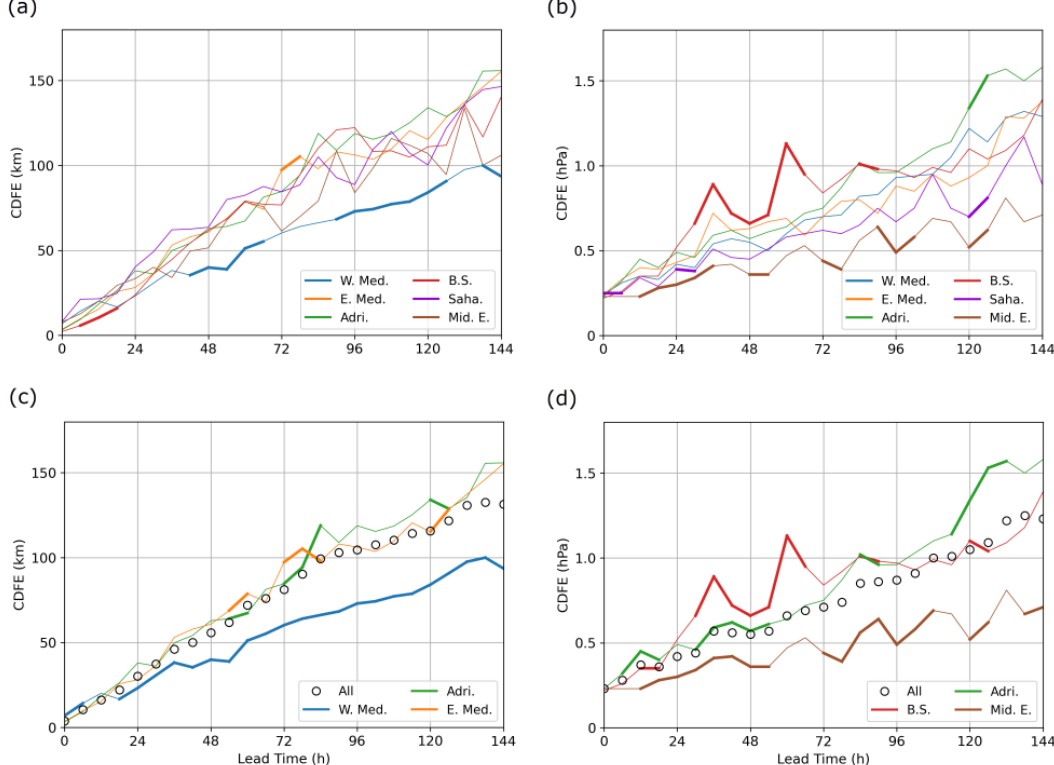

**Figure 8.** Differences in the predictability between the regions defined in Fig. 1b, for (a) and (c) the total track error (TTE), for (b) and (d) the MSLP error (MSLPE). Results appear in thick lines when they are significantly different from every other categories. The CDFE score computed from the complete data set is represented by the black circles.

complete data set (black circles). West Mediterranean cyclones are for their part significantly better predicted from 18 h lead

time and beyond, the difference with the worst categories reaching more than 50 km at 144 h.

In Fig. 8b, the differences in the predictability considering the MSLPE are presented for the complete set of six regions. The significance of the results is less pronounced than for the location, but regional differences can still be observed, in particular between the best and the poorest predicted categories (Fig. 8d). The Middle East is the region in which the intensity of cyclones is the best predicted at each lead time. A clear diurnal cycle is observed in this region, with local CDFE maxima at 42 h, 66 h,

90 h, 108 h and 132 h lead time, corresponding to local times of 3 p.m. to 9 p.m. While the coarse temporal resolution of 6 h does not allow a precise timing of this behaviour, it seems that cyclones in this region are experiencing greater errors during the warm part of the day. The cyclones in the Black Sea are for their part the poorest predicted in the firsts 72 h, and once again a diurnal cycle is observed with two pronounced maxima at 36 h and 42 h, corresponding to the beginning of the afternoon in this region. Trigo et al. (2002) already identified diurnal cycles in summer cyclones developing over northern Africa, the

Iberian Peninsula, the Black Sea, or over the Middle East. The maximum intensity was reached during the afternoon and





cyclolysis generally occurred in the early morning. The reason for the diurnal cycle of errors shown here could be linked with the representation of the convective processes, often occurring during the afternoons of summer days.

## 5.2 Seasonal differences

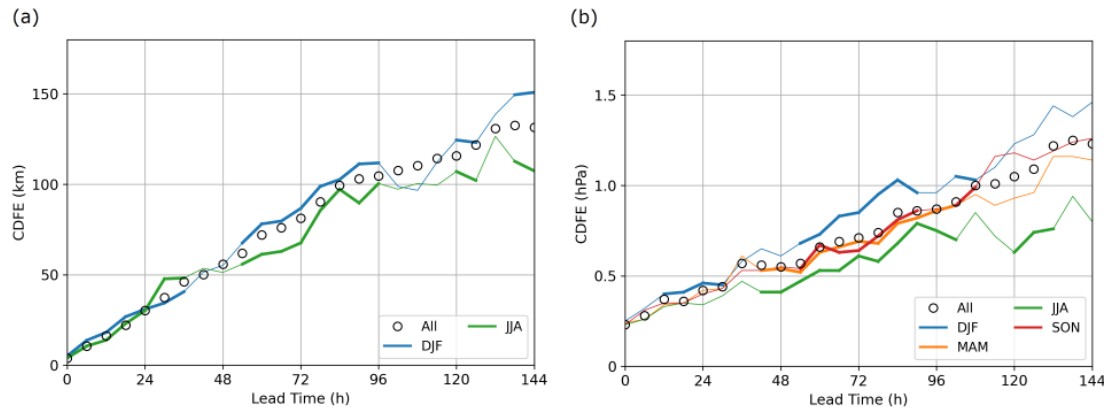

**Figure 9.** Differences in the predictability between the seasons (*e.g.* DJF stands for December January February), for (a) the total track error (TTE) and (b) the MSLP error (MSLPE). Results appear in thick lines when they are significantly different from every other categories. The CDFE score computed from the complete data set is represented by the black circles.

Another possible categorisation of Mediterranean cyclone is based on the seasonality. As previously, Fig. 9a presents the
CDFE score for the TTE and Fig. 9b for the MSLPE. Only the categories significantly different from the others are represented. In terms of errors of location, winter cyclones (December-January-February) are generally less well predicted than summer ones (June-July-August). The results are statistically significant for these two extreme seasons, but the difference remains under 25 km before 132 h lead time. Consequently, the season in which the cyclone occurs does not appear to be determinant in the predictability of its location.
Differences are more pronounced in the intensity, and results are statistically significant from 48 h to 102 h lead time for every seasons. Two major observations can be made. Firstly, the CDFE score in the autumn and spring follows the general MSLPE of all Mediterranean cyclones (black circles). Errors are greater than average in winter and smaller than average in summer. Secondly, it is noticeable that winter and summer cyclones are significantly different from each other at every lead time (not shown), with an increasing difference between the two categories from 0.1 hPa at 24 h to more than 1 hPa at 144 h
lead time.




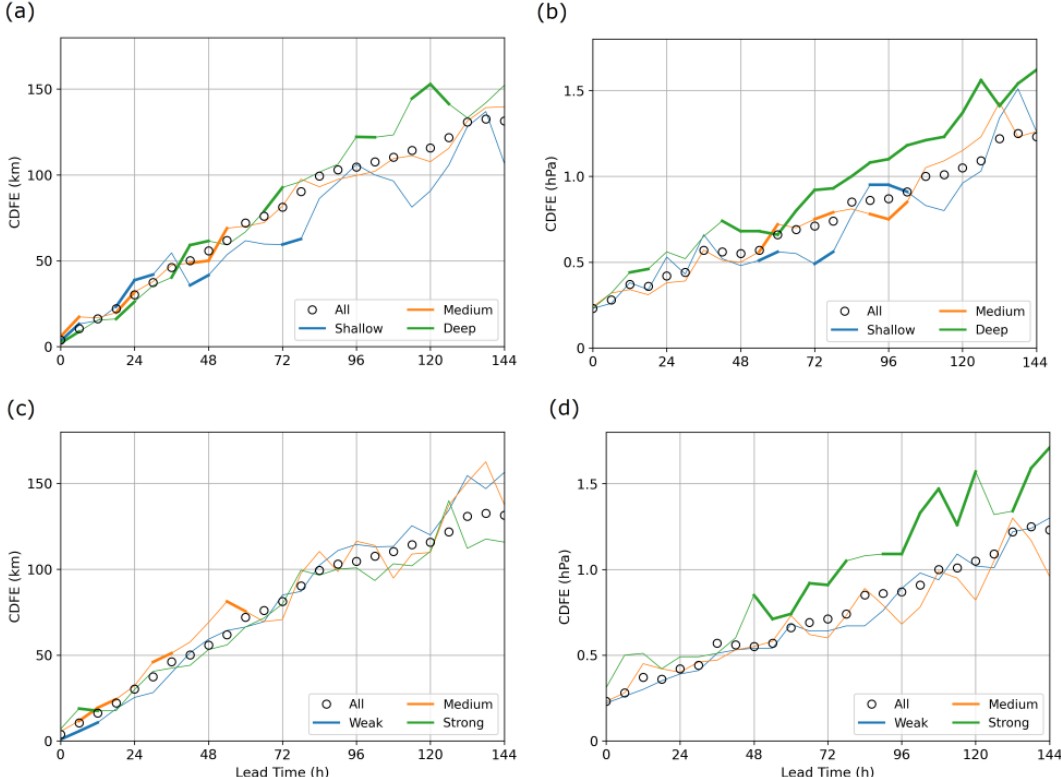

**Figure 10.** (a) and (b): Differences in the predictability between 3 intensity-based categories of Mediterranean cyclones following their minimum MSLP, namely the 10 % shallowest, the 10 % around median intensity and the 10 % deepest. (c) and (d): Same as (a) and (b) but based on the deepening rate, defined as the pressure difference between the MSLP at the time of maximum intensity and 12 hours before. (a) and (c): Results for the total track error (TTE). (b) and (d): Results for the MSLP error (MSLPE). Results appear in thick lines when they are different significantly from every other categories. The CDFE score computed from the complete data set is represented by the black circles.

## 5.3 Intensity classes

Differences in the predictability for different intensity-based categories are shown in Fig. 10a. They are weakly significant for the TTE, even if it seems that the deeper the cyclone, the poorer the predictability in location for lead times greater than 72 h. The predictability of the cyclone location is also independent of the deepening rate (Fig. 10.c).

In terms of MSLPE, the deep cyclones are clearly the poorest predicted after 72 h lead time (Fig. 10b). In contrast, shallow cyclones are not necessarily better predicted than the medium category, and the difference is not always significant between these two categories. Same conclusions can be drawn from the deepening rate (Fig. 10d), where rapid intensification cyclones strike out with intensity errors significantly greater than in the other categories after 72 h lead time. It is in agreement with Pantillon et al. (2017) and Pirret et al. (2017), who both showed a poor prediction of the intensity of the severe European



storms they investigated. However, it should be noted that on average, the intensity of deep storms in our data set is slightly
over-predicted from day 4.5 onward, while it is slightly under-predicted in these two previous studies. This difference could find
an explanation in the samples of studied cases, as Pantillon et al. (2017) and Pirret et al. (2017) find a slight under-prediction
in a data set of 25 and 60 extreme North Atlantic storms, respectively, while 280 less extreme Mediterranean cyclones are
represented here in the deep cyclones category.

To summarise, the predictability is significantly poorer in terms of MSLPE for deep and rapid-intensification cyclones, from
72 h lead time and beyond. As seen in Section 3.3, these poorly predicted cyclones tend to form during the cold part of the
year (Fig. 3b), in agreement with the poorest predictability of winter cyclones shown in the previous part. They are also mainly
located in the West Mediterranean and in the Adriatic, with a direct influence of the Atlantic (see Fig. 3a).

## 5.4   Difference between velocity classes

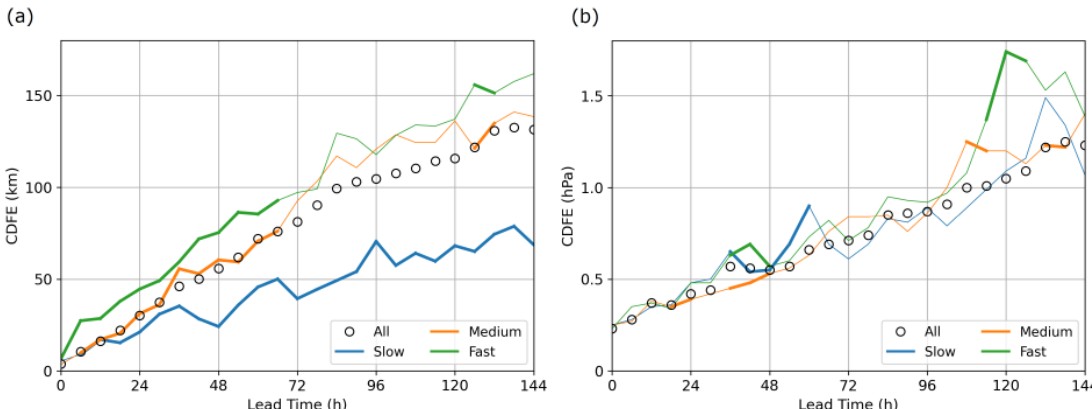

**Figure 11.** Differences in the predictability between 3 velocity-based categories, namely the 10 % slowest, the 10 % around median velocity
and the 10 % fastest. (a) for the total track error (TTE) and (b) for the MSLP error (MSLPE). Results appear in thick lines when they are
different significantly from every other category. The CDFE score computed from the complete data set is represented by the black circles.

It has been demonstrated based on Fig. 4 that different velocity-based categories of Mediterranean cyclones have different
spatial distributions. It is consequently expected that differences will also appear in the predictability, which varies between
regions (Fig. 8). Figure 11a presents the CDFE metric for the velocity-based categorisation of cyclones examined in Section 3.4.
The link between the velocity of the cyclone and the predictability of its location is remarkable, and differences are statistically
significant from the beginning of the forecast until 66 h lead time: the faster the cyclone, the poorer the predictability. The slow
cyclones are clearly better predicted than any other category, especially for lead times longer than 36 h. The difference with
the two other classes is statically significant and increases with lead time, reaching almost 100 km after 6 days of forecast.
The particularly good predictability of these slow cyclones has to be linked with the unique spatial distribution highlighted in



Fig. 4c. Indeed, these quasi-stationary lows are in a vast majority concentrated in the Gulf of Genoa in the West Mediterranean, which is the region where the cyclone location is the best predicted (see Fig. 8).

Unlike for the location, the velocity of the cyclones does not play an important role in the predictability of the intensity (Fig. 11b). At lead times longer than 114 h, the fastest cyclones are the worst predicted, but the difference with the other categories is too small to build any robust conclusion.

## 6    Summary and conclusions

The predictability of extratropical cyclones can be highly variable from a case to another. Here, an approach based on the use
of both reanalysis and ensemble reforecasts with a constant model configuration over 20 years makes it possible to investigate the predictability of Mediterranean cyclones in a systematic framework.

Cyclones are first tracked in the ERA5 reanalysis, providing a large reference data set of 2853 cyclones over the 2000-2021 period. Their spatial distribution is in agreement with most of the previous climatological studies, confirming the inhomogeneity in the distribution of cyclones in the Mediterranean. Six preferred regions concentrating 63 % of the data set are identified,
the Gulf of Genoa being the main hotspot in the region. Comparatively to previous studies, a high density of cyclones is found in the Adriatic. A clear seasonal cycle is highlighted, with a higher occurrence during the cold part of the year. The cold season is also more favourable to the development of intense cyclones, which mainly occur in the West Mediterranean, in the Adriatic, and in the north-western parts of the Black Sea.

Reference cyclones are then tracked in the homogeneous set of ensemble reforecasts for the same period. The predictability
is evaluated in terms of errors in both cyclone location and intensity. Comparable magnitudes between mean error and spread indicate a reasonably good reliability of the IFS ensemble reforecasts for Mediterranean cyclones. A slight over-dispersion of the ensemble can however be observed at every lead time, whether in the location or in the intensity. It should also be noted that while the ensemble spread is slightly greater than errors for most of the cases, some cyclones remain very poorly predicted with a mean error that can be more than 4 times greater than the ensemble spread.

The errors are summarized for the large number of cases by generalizing the CRPS probabilistic score to the newly-defined CDFE score based on the error distribution. Considering the entire set of cyclones, it is shown that the median location error grows at two different rates with increasing lead time. In the first 78 h, the error grows at a constant rate of 40 km per day, comparable to the one found in (Picornell et al., 2011) for Mediterranean cyclones. The growth rate is however two times smaller from 84 h lead time and beyond. This behavior is attributed to the saturation of errors and to limitation inherent to the
verification of tracks against the reference. In terms of intensity error, a very weak and almost constant bias of -0.2 hPa per day is detected, indicating a slight overestimation of the intensity in forecast cyclones, in agreement with Froude et al. (2007b) for North Atlantic cyclones. This result should be regarded with some caution, as reforecasts are not compared with observational data but with reanalysis data, which may underestimate the actual cyclone intensity.

Looking at different categories of Mediterranean cyclones allows to determine several factors contributing to a better or
poorer predictability. In terms of cyclone location, the velocity appears to be the key factor. In particular, the slowest Mediter-



ranean cyclones, which are mainly located in the Gulf of Genoa, are much better predicted than any other category, at every lead time. The impact of such quasi-stationary cyclones can be important by causing large amounts of accumulated precipitation in a same area. The predictive skill in their location is therefore important. For their part, the location of the fastest cyclones is relatively poorly predicted in the first 66 h lead time. To the authors' best knowledge, it is the first time that a link between the cyclone's velocity and predictability is highlighted.

Two factors leading to differences in the predictability of the cyclone intensity are clearly established. First, errors in the intensity of deep cyclones are greater than in any other category after 3 days of forecast. This is also observed for the deepening rate, where the prediction of rapid-intensification cyclones is the poorest. It is in agreement with Froude et al. (2007a), who have shown a relatively poorer predictability for intense cyclones in the Northern Hemisphere. A second important factor in the prediction of the intensity is the season in which the cyclone occurs. Winter cyclogeneses are indeed more poorly predicted than summer ones, and the difference between these two seasonal categories increases with lead time. In fact, the two factors are strongly related, as the deepest Mediterranean cyclones occur almost exclusively during the cold part of the year. The forecast skill for the intensity of those strong winter cyclones is important, as some of them account for the most destructive windstorms in the Mediterranean (*e.g.* storm Klaus: Liberato et al., 2011). Froude et al. (2007a, b) suggested that errors in the intensity of deep cyclones could originate from an incorrect representation in their vertical structure, as the vertical tilt is known to play a major role in storm development. This hypothesis has to be verified in a systematic way for the Mediterranean.

In this study, the predictability has been quantified in a systematic framework for several categories of Mediterranean cyclones. The cyclone velocity, its intensity, the season and the region in which the cyclone occurs are all playing a role. Further investigations could focus on physical processes responsible for the loss of predictability. In particular, the quantitative importance of baroclinic and diabatic processes in the poor predictability of deep Mediterranean cyclones should be addressed. Indeed, both the representation of latent heat release in the firsts forecast hours and the location of Rossby wave breaking at high lead times (several days) may be responsible to the loss of predictability of Mediterranean cyclones. It could also be interesting to find a physical explanation to the remarkable good predictability of the shallow cyclones in the West Mediterranean.

*Code and data availability.* The tracking algorithms are available in the open-source TRAJECT software https://github.com/UMR-CNRM/Traject. The ERA5 reanalysis is available through the Climate Data Store https://cds.climate.copernicus.eu/. The IFS reforecasts are available on the MARS Catalogue https://apps.ecmwf.int/mars-catalogue/ (restricted access). Cyclones tracks are available on request.

*Author contributions.* BD performed the analysis and wrote the initial draft. FP retrieved the reanalysis and reforecast data. MP developed the tracking algorithms. TR provided expertise on statistical parts of this manuscript and LD on the evaluation of the ensemble prediction system. All authors participated in designing the study and preparing the final draft of the manuscript.



*Competing interests.* The authors declare that they have no conflict of interest

*Acknowledgements.* This work was funded by Région Occitanie and Météo-France through project PREVIMED. Is also received support from the French National Research Agency under Grant ANR-21CE01-0002 and from COST Action CA19109 "MedCyclones: European Network for Mediterranean Cyclones in weather and climate". The authors thank ECMWF and Copernicus for making the ERA5 reanalysis and the reforecasts available.



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
