# Peer review of "Systematic evaluation of the predictability of different Mediterranean cyclone categories"

_EGUsphere, 2024_

## Referee Comment (RC2)

**Review of '*What determines the predictability of a Mediterranean cyclone?*' by Benjamin Doiteau, Florian Pantillon, Matthieu Plu, Laurent Descamps, and Thomas Rieutord**

The aim of this paper is to investigate the predictability of Mediterranean cyclones when conditioning on different categories, namely region, season, cyclone intensity and motion speed. In a first step, reference tracks are identified in ECMWF's ERA5 data set and results are presented in section 3, documenting the climatology of that data set and discussing commonalities with and differences to previous studies. The reference cyclones are then tracked in ECMWF's 11-member ensemble reforecasts to compute location and intensity errors as well as different measures for means, median and spread. These are discussed together with an assessment of forecast reliability in section 4. Finally, practical predictability is evaluated condition on the categories mentioned before, using a novel approach that defines CDFE scores. The underlying idea stems from the formulation of the CRPS and integrates the area between an error CDF and the Heaviside function with the step at 0.

The results show that the predictability of cyclone location is mainly determined by cyclone motion speed, while the intensity and deepening rate are found as the two key factors that determine how good the intensity is predicted. These results are an important first step to identifying and better understanding the main drivers for predictability of Mediterranean cyclones. However, I strongly suggest to rephrase the title as there are many more factors that determine predictability than the four aspects studied here. The paper is well structured and written and the results are generally clear. However, there are two main concerns: First, the cyclones comprise a broad range of storm natures (extra-tropical to tropical-like), which may have different predictability associated with different natures, but this is not addressed at all. Second, two different tracking algorithms are applied to the reference and reforecast data, which may unnecessarily affect the calculation of errors, and hence the results of the predictability study.

I recommend the paper for publication after consideration of the general comments below. Specific comments are also provided to help improve the paper.

**General comments**

1. Title: I find the words 'What determines' too open for what is covered in the paper. The study looks at differences in predictability based on the region, season, intensity and motion speed, respectively. And these factors are definitely important. However, the word 'what' could also refer to differences in precursors, the influence of the environment or physical processes (as mentioned in the outlook paragraph in the summary and conclusions section). I therefore suggest specifying the title so that it better reflects what is being investigated.
2. As described in the introduction section, there is a certain range in the nature of cyclones in the Mediterranean region, spanning from deep extratropical cyclones to tropical-like Medicanes. Therefore, it can be assumed that the predictability also depends on the nature of the storm, as different processes may play a role during formation. However, the paper makes no distinction on this point, which is why it is questionable to what extent the results are comparable to other regions (the paper often compares with the Atlantic or the whole northern hemisphere), as they may cover a different range on the spectrum of storm natures with different frequency

distributions. If examining this aspect in the reforecasts is considered to be beyond the scope, I yet recommend to add some analysis and discussion in this regard for the reference tracks, to at least document the distribution of storm natures for your data set. There are different approaches to this, but one way would be to calculate CPS metrics, for which code would be available, for example here: https://github.com/fredericferry/era5_cps_diagram

3. Apart from this additional analysis, the term 'Mediterranean cyclone' should be defined more precisely in terms of what range of cyclone natures is spanned. Also, please check the entire manuscript to see where the term 'cyclone' can be specified more precisely in order to improve clarity on this point.

4. You introduce the two 'types' of predictability (practical vs intrinsic) in the introduction, but never use these terms thereafter again. Since your study is based on forecasts from the IFS model, I suggest to clearly state once that the predictability you are assessing is of the type 'practical', and then say: 'hereafter referred to as 'predictability' for simplicity'.

5. Cyclone detection in the Ayrault algortihm: According to lines 119-120, the resulting track may be a mixture of $\zeta$-based and MSLP-based points. Why are tracks created based on such an inconsistent use of variables? Since the study uses error distributions of cyclone location (and intensity) as the basis to investigate predictability, I consider this point to be crucial with regard to all conclusions based on it. Could you check how different reference tracks are, when solely using $\zeta$-based track points?

6. Why is a different tracking method used for the reforecasts? For me, it would be more obvious to use the same method for both data sets. In lines 158-160, you say that you refrained from applying the Ayrault algorithm to the reforecasts. But why did you not apply the VDG algorithm to both datasets (i.e., reforecasts and ERA5), to keep it consistent? Is there any particular reason? One downside of this inconsistency is that the 'data and methods' section becomes (in my opinion unnecessarily) longer.

7. As the word "velocity" can be used and understood in different contexts, I suggest using 'motion speed' throughout instead.

8. I suggest to harmonise the titles of sections 5.1-5.4. Either use 'Differences in X' or 'X categories'.

**Specific comments**

l. 1: 'in a densely populated area' is very unspecific. Rephrase so that the geographical location does not have to be implicitly deduced from the term "Mediterranean cyclones".

l. 5–6: 'this region': Make sure to have clearly stated, either here or in the first sentence, which region you are referring to.

l. 5–6: Incorrect name: ECMWF is the "European Centre for Medium-Range Weather Forecasts".

l. 16-17: "opens the way to identify the key processes…" is what a reader may expect when reading the title, but it is not addressed in the present manuscript. Therefore, I strongly recommend to specify the title as mentioned in one of the general comments.

l. 19: The first sentence of the introduction is very similar to the first sentence of the abstract. However, while Mediterranean cyclones are described as a 'component of the climate' in the abstract, extratropical cyclones are introduced as 'component of weather patterns' in the former. Is there a particular reason why they are introduced in conjunction with different temporal (and spatial) scales? If so, please clarify, otherwiese I would recommend to use consistent wording.

l. 24: Replace 'life cycle' by 'life time'.

l.24: The previous paragraph specifcally addresses extratropical cyclones. Here you make a comparison between cyclones in the Mediterannean and other basins, not mentioning the storm's nature. For the statements you make, are you comparing extratropical / subtropical / hybrid / tropical(-like) cyclones in the Mediterannean and other basins? Please clarify.

l. 24-36: Only with the introduction of medicanes in l. 36 you begin to discuss cyclone nature for the Mediterannean. Please make sure that a reader knows which type (or range on the spectrum) you are addressing in these lines.

l. 47: Note that Baumgart et al. (2019) did not investigate error growth in the specific context of cyclogenesis and cyclone prediction, respectively. Consider this in the text.

l. 53: Replace 'hardly' by 'rarely'

l. 54: The comparison seems to be inappropriate. The purpose of the ensemble is to quantify the uncertainty beyond a purely deterministic prediction, and hence provides an additional perspective. The word 'robuster' (should be 'more robust') only makes sense if one compares a single forecast with an ensemble mean.

l. 56: There are more intensity categories below the hurricane intensity threshold. Therefore, bringing together the word 'hurricane' and 'genesis' seems a bit odd. I suggest to say 'tropical cyclone genesis' instead.

l. 64: Add comma before 'but'. Also, I suggest to think of (and write about) Medicanes as a subset of the 'Mediterranean cyclones', instead of emphasising the lack of representativeness.

l. 66: What nature (or specturm of cyclones) of cyclones are you referring to?

l. 69: Replace 'firsts' by 'first'.

l. 70: Replace 'systematical' by 'systematic', and remove 'as objects'.

l. 71: What type of vorticity? Relative vorticity?

l. 72: Replace 'degree by 'degrees'.

l. 74: Replace 'and this' by ', which'.

l. 76: Since this paragraph reports on results from previous studies, mainly addressing model deficiencies ('incorrect representation','systematic slow bias', etc.), it's probably worth

specifiying the the predictability here and in l. 69 as 'practical predictability'. But I leave it to the authors to decide.

l. 79: What is meant by 'with a homgenous configuration'. Please clarify.

l. 79-80: Replace 'robust statistical signals' by 'statistically robust signals'.

l.81: In the main text, you have not referred to specific ensemble prediction system yet, so either remove 'the' before 'ensemble' or state which one you are going to use.

l. 84: Add 'cyclone' before 'tracking'.

l. 91: Add a hyphen between 'Medium' and 'Range'.

l. 91-92: Remove 'hereafter'.

l. 95: If you have used a regular lat-lon grid, replace 'with 0.25° horizontal resolution' by 'on a 0.25°x0.25° horizontal grid', as it is relevant technical information (for the purpose of reproducing the method/results).

l. 98: Replace 'need' by 'step'.

l. 100: 'Coarser' than what? Can you give an approximate number?

l. 101: Replace 'required a specific tuning' by 'had to be adapted', 'Indeed' by 'As stated before, 'with' by 'have' and 'life cycles' by 'life time'. Also, add 'those' before 'in the'.

l. 102: Add a comma before 'and'.

l. 103: Add 're-' before tuned.

l. 104: Add 'to' before 'operate'.

l. 107: 'in' → 'into'.

l. 108: Is there any reason why you have chosen these characteristic lengths? If so, please add a sentence stating it. Maybe also consider adding a sentence explaining the reason for the smoothing (i.e., why you apply a moving average). And since you consider a broad range of cyclones (large extratropical to smaller tropical-like), did you test whether/how your choice works equally well when tracking different cyclone types?

l. 111: To avoid confusion with the two levels used for winds, add 'at 850 hPa' after 'smoothed field'.

l. 112: When retaining the strongest maxima, is this evaluated for each grid point separately on the full set of maxima identified with then 300km radius or does the dropping of maxima affect the search at grid points in the neighbourhood? Depending on the procedure, revise the description accordingly to ease reproducability.

l. 113: 'on' → 'over', and add 'in a given reforecast' after 'time steps'.

l. 114: Replace 'three steps' by 'three-step'.

l. 115: Add 'for time t+1' after positions, and remove the word 'for'.

l. 115: How large is the neighbourhood? Please state it clearly in the text.

l. 116: The term 'by taking into account' is not a precise methodological description that would allow for reproducibility. Please clarify how that distance value variation (which I assume is variance?) is used in the quality criterion, to allow the reader understand how the final position is determined.

l. 119: Replace 'in' by 'within', and add a hyphen between '3°' and 'square'.

l. 121: Replace the second 'of' by 'if'.

l. 121-124: In line 120 you describe that it can happen that for some $\zeta$-based track points there is no matching MSLP points identified. In such cases 'the $\zeta$ amximum remains the track point'. How can you check validation criteria based on MSLP, when there are track points, for which no match was found? What MSLP value is associated with such points instead?

l. 123: Replace 'life cycle' by 'life time'.

l. 124: Swap words: 'secondary local' $\rightarrow$ 'local secondary', and replace 'northern' by 'Northern'.

l. 125: Replace 'that enter into' by 'entering'.

l. 126: By 'areas', are you referring to the areas that will be defined later? If so, it can't be used here because it has not been introduced yet. Maybe remove the word.

l. 127: Is 'the Mediterranean-adapted version of the Ayrault (1998) algorithm' referring to the algorithm you are using in your work or another algorithm? To me, it sounds like it describes the adaptations you did to the Ayrault (1998) algorithm. Please clarify. And replace 'slight' by 'slightly'.

l. 129: 'distribution' $\rightarrow$ 'distributions'

l. 131: What does 'in the highest confidence level' mean? Please clarify.

l. 134: Since the reference tracks are also used as part of the predictability study, I would replace 'predictability study' by 'reforecast tracks', to be consistent with the title of subsection 2.1.

l. 135: There is no need for a period per se, you could just run a single reforecast. Therefore, I suggest to replace 'on a historical period spanning typically several decades' by 'starting from historical intial conditions'.

l. 136-137: Given that you already state in the previous sentence, that 'reforecasts are forecasts …', I don't see the need for this sentence and would hence remove it. Together with the previous comment, it should be clear that they are run from past initial conditions

(without taking into account observations afterwards). If you remove, also remove 'therefore' in line 137.

l. 146-147: 'life cycle' → 'life time'

l. 147: Remove 'for'.

l. 150: Replace 'of' by 'for'.

l. 150-151: Having read the last two sentences of this paragraph several times, I am not sure I'm getting your point. Please revise it to make it clearer.

l. 152: Same as for the title of section 2.3: 'predictability study' → 'reforecast tracks'

l. 154: Given that you applied a different tracking method to the reforecasts, it is worth mentioning why already after the first sentence of this paragraph? I suggest to move lines 158-160 after '… tracking of tropical cyclones.' (in line 154).

l. 158: Replace 'track predicted cyclones' by 'tracks predicted for cyclones'.

l. 159: To make clearer that the 'additional step for matching the cyclones' is needed for cyclones (first identified at lead times>0) and becomes both more important and challenging as lead time increases, I would replace 'the cyclones' by 'forecasted and observed cyclones, which are first identified at lead times greater 0'. I would remove 'subjectivity' as there are several of objective methods in the literature addressing exactly this issue.

l. 161: Add 'time' after 'initialisation'. Also, here and were necessary throughout the paper, replace 'ζ maximum' by 'ζ-maximum'.

l.162-163: The word 'combination' is used twice, replace ones.

l. 163: 'position' → 'positions'.

l. 167: Replace 'and here set at 0.5, and δt the time step' by 'of the forecast δt, but here set to 0.5'.

l. 170: Replace 'using' by 'within' and 'around' by 'on'.

l. 175: Add the statement that you did not apply the other two requirements as in the Ayrault algortihm (i.e., a minimum track length of at least 24h and a minimum intensity of 1005 hPa that has to be reached), and maybe say why. Probably, because you know that there was a cyclone observed based on ERA5 but you don't want to restrict your ensemble reforecasts too much. Correct?

l. 176-177: A very important information that I would suggest to tell the reader before the VDG algortihm is described in detail! I'm sure it will help avoid some questions that may arise.

l. 181-182: remove 'with each other, whether in location or in intensity', as it is a general condition.

l. 183: 'for each cyclone track' is ambiguous as there are observed and (for a unique cyclone usually multiple consecutive) forecast tracks. I suggest to remove it and replace 'defined' by 'calculated'. Also add 'each' after 'of' and remove 's' from 'members'.

l. 184: 'the cyclone life cycle' → 'a cyclone's life time'.

l. 184-185: Is there a particular reason why you chose to calculate spread as the mean of pairwise differences rather than the standard approach based on the ensemble mean? (By ensemble mean one would of course consider only the subset of members that have a cyclone at time t.)

l. 188-189: Remove the four words 'respectively'.

l. 190: 'defined' → 'calculated'

l. 191: remove 'an'. Also add 'forecast' after 'for each'. 'life cycle' → 'life time'.

l. 195-196: Remove the two words 'respectively'.

l. 204: remove 'resp.'.

l. 211: Replace 'CDFE($\tau$)' by 'CDFE($F_\tau$)'.

l. 213: Add '.' after 'applied'.

l. 213-214: Replace 'A higher CDFE (respectively the smaller) indicates a poorer (respectively a better)' by 'A higher (smaller) CDFE indicates a poorer (better)'.

l. 215: With your significance test, You can only test for either similarity or difference at a time, as you would need to define a different null hypothesis otherwise. Remove 'respectively', and add 's' after 'CDF' (plural).

l. 215: Replace '(respectively different)' by 'or not'.

l. 221: Add 'the' before 'toponymos'.

Figure 1: The spatial distribution is only one aspect. Isn't it the (relative?) frequency of occurrence what you are showing? This also includes the temporal aspect. Also, remove 'the' in the last sentence. For (b), add a note that the informs about the nonlinear shading levels.

l. 223-249: Following the previous comment, I would suggest to replace 'spatial distribution' by 'relative frequency'. If you change it, make sure to also change it in line 220.

l. 224: Add a comma after 'period'.

l. 228-229: Add commas before desgined and after areas.

l. 231: The reference (Trigo et al. 2002) for this statement made here is only mentioned a couple of lines later. Can you include it here already?

l. 247: 'at' → 'of'

l. 248-249: Since it is not found in most of the other studies, do you have an idea of potential (maybe methodological?) reasons for this? If so, it would be worth to mention. Having read lines 125-126 again, I wonder how this condition is actually defined. Does 'enter into' refer to the domain enclosed by the box (as shown in Fig.1b) or to the entire sea basin? If the former, does it matter whether a cyclones enters the box over land vs over sea? Please clarify.

Figure 2: Replace 'on' by 'over'. What does 'in their mature phase' mean? Is Fig. 2 only based on the most intense fragements of your reference tracks? If so, it should be stated clearly how this is filtered.

l. 267: 'rate' → 'rates'

l. 267-268: Can you add a reference for the link to the Atlantic?

l. 269-279: Here you discuss the depth of cyclones, but it is never stated in the main text (but only in the caption of Fig. 3) how these catagories are defined. Please clarify either here or in the 'data and methods' section.

Figure 3: As before, consider replacing 'spatial distirbution' by 'Relative frequency'. I suggest to move the definitions of the categories to the main text and then add soemthing along the lines 'See the main text for details on the category definitions.'. For (a), add a note that the informs about the nonlinear shading levels.

l. 279: As suggested in the general comments, I would replace 'velocity' by 'motion speed', and remove 'of Mediterranean cyclones' to stay consistent with the previous subsection titles.

l. 280: Why have you chosen the median and not the mean? Is it because the distribution is so skewed? Probably worth to mention. How large are the differences to the mean? Also, 'life cycle' → 'life time'.

l. 281: Replace 'generally move from the west to the east' by 'move eastward on average' as there are cases where it moves westward (e.g. Medicane Daniel). Can you also add a statement on the meridonal component?

l. 282: Remove 'the' before 'twice'.

l. 283 & l. 285: As is section 3.1, replace 'spatial distribution' by 'relative frequency'.

Figure 4: As with Fig. 1b+3a, replace 'spatial distribution' by 'relative frequency'. Add a note that the informs about the nonlinear shading levels.

l.284-285: 'change' → 'changes', and replace 'from one velocity-based category to another highlights' by 'between the motion speed-based categories highlight'.

l. 296-297: As you do not distinguish between different stages of a cyclones life time in your study, I would refrain from linking different motion speed-categories to differences in processes of cyclogenesis. It may be that a cyclone forms in one part of a region but then moves into another part of the same region, while changing motion speed. Therefore, here

and throughout the paper, I recommend to not speculate on cyclogenesis, if it is not specifically analysed.

l. 302: I suggest to use a more concise title for 4.1: 'Location and intensity errors'.

l. 303: Remove second 'in', and 'Fig.' → 'Figure'.

l. 304: ', for the all' → ' for the entire'.

l. 305: It is not clear to me which hypothesis you want to test, which is why the topic of significance is not appropriate here. Presumably you want to say that the large number leads to more stable results!?

l. 306: Add 'approximately' before 'linearly', and 'on' → 'in'.

l. 308: 'until' → 'up to'.

l. 309: '3 day' → '72 h', to be consistent with the x-axis.

l. 310: 'not fully linear': Instead, I suggest to describe how it actually is, namely 'slower-than linear'.

l. 311: As you talk about the TTE growth rate x in the previous part of the sentence, 'it increases' is misleading, as the growth rate is actually decreasing. If you repplace 'it' by 'TTE', it should be fine.

l. 312 & 316: Maybe replace 'processes' by 'reasons'.

l. 314-315: I find it difficult to understand what point you are trying to make here. Please clarify and rephrase the sentence. Also, do not use 'early lead times' and 'from a few hours', as it is not clear what it means. Use 'short(er)' or 'long(er) lead times' instead. And change 'i.e' to 'i.e.,'.

l. 316: 'deals with' → 'has to do with'.

l. 316-317: In principle, a forecast can predict anything, but what we want is that it ultimately converges to climatology. Since an ensemble forecast is considered here, 'a random climatological state …' is not adequate wording. Maybe say: 'For long enough lead times, an ensemble forecast should ultimately converge to the climatological distribution'.

l. 318: Remove 'a' and change 'value' to 'values', as mean and median may not necessarily converge to the same value if the climatological distribution is skewed.

l. 321: What maximal growth rate do you get, when calculating it over the first 48 h only, as Picornell et al. (2011) did?

l. 322: A comparison with the "whole Northern Hemisphere" means comparing to a broader spectrum of cyclone natures (e.g. including true tropical cyclones), and thus different error characteristics (i.e., distributions). If you want to keep this comparison, you should point out

this important difference. Given the much coarser resolution of that study, I wonder whether it would be better not comparing to it.

l. 327: Remove 'In our case,'.

l. 328: 'from' → 'at'. Also, replace 'are slightly late' by 'have slow propagation speed bias'.

l. 329: 'in average' → 'on average'. Also, add 'in the IFS model' after 'forecast cyclones'.

l. 330: Remove 'For their part,'.

l. 332: Add 'ly' after 'previous', and a comma after 'CTE'.

l. 333: 'shift' → 'bias'

Figure 5: Change 'compared to ERA5 in function of the lead time' to 'relative to ERA5 as a function of lead time'. Replace 'first and third' by 'first to third'. Also, replace 'black lines' by 'black whiskers'.

l. 335: 'when looking at the mean': What is 'mean' referring to here? The ensemble mean? The mean over all lead times considered? Please clarify. If the ensemble mean, the lead time reference is missing.

l. 336: Add 'error' before 'distribution', and remove 'around the reference' as you can simple say that the MSLP errors are centered around zero.

l. 337: '3 days' → '72 h'

l. 338-339: Is there a specific reason why you are reporting the IQR at 72h lead time? If you want to include in the text, maybe describe how the IQR increases with lead time.

l. 339: Same as above: The question is whether it is reasonable to compare to this deviating range (i.e., frequency distribution) of northern hemispheric cyclone natures.

l. 340: 'indicates' → 'indicate'.

l. 343: 'event' is rather unspecific. Please clarify what it means. What about using 'cyclone-base time combination' (or 'pair' instead of 'combination')? Admittedly, it sounds more complex, but describes it more precisely.

Figure 6: Suggestion: Replace the first sentence' by 'Spread-skill relationship at 72h lead time with blue shading representing the number of cyclone-base times populating each bin.'. In the last sentence, replace 'event' as in the previous comment, and change 'in each half part' to 'above and below the diagonal, respectively'.

l. 345-346: Remove the two words 'respectively'.

l. 348: Replace the 'in's by brackets around 'Fig. 6x', and replace 'in both cases' by ' for both aspects'.

l. 349: 'trend' → 'relationship', 'reliable' → 'calibrated'.

l. 350: Since you are make these statements for both plots at once, add about before '60%' as it is '58%' for MSLP. And as above, be more specific and replace 'event'.

l. 353: Suggestion: 'is slightly over-dispersive for most of the cyclone cases' → 'tends to be over-dispersive in most forecasts', and 'remain very poorly predicted' → 'are totally off'.

l. 355: Add 'really' before 'observed', and replace 'the different' by 'other' or 'all'.

l. 357: Add a comma after 'section'.

l. 359-360: Change 'in the spatial distribution, the seasonality' to 'depending on the region, the season'.

l. 362: Replace 'Note that the CDFE having the same unit as the variable considered, the' by 'It should be noted that the CDFE has the same unit as the variable considered. The'

l. 363: Remove two times 'respectively the', and move '(smaller)' after 'greater' in line 362.

Figure 7: Move 'at 72 h lead time' after '(b)'.

l. 364: 'On' → 'in'

l. 365: Add 'the shape of' before and 'error' the word 'its'. Also, 'is closer to' → 'better resembles'.

l. 371: 'sections' → 'subsections', remove 'the' and put 'score' in the plural. Also change 'is' to 'are'.

l. 372: Add 'considered' after 'duration'

l. 374: 'spatial distribution' → 'relative frequencies'

Figure 8+9+10+11: What does 'from every other categories' mean? Do you test against every other category separately and then mark the parts where all intersect? Or do you test against all other (i.e., remaining) categories combined? If the latter, replace 'every' by 'all'. In any case, this should be clearly explained here and in the text. Also, what is plotted when a single lead time has a singificant difference (i.e., with the previous and following lead times being insignificant). Is it a circle or a point, or is it possible that nothing is drawn because there is no other point with which a line could be constructed? Also, replace 'The CDFE score' by 'CDFE scores', and change 'is' to 'are'.

Figure 8: Fig. 8c+d are not needed as they do not provide additional information. Another reason to remove them is that the significant parts change as they are probably now derived testing against only two of other regions, which is a bit confusing (as it is not mentioned in the text). Also, add the black circles in panels (a) and (b).

l. 377: Add 'in terms of cyclone track' after 'predicted'.

l. 378 As for the captions of Fig. 8-11, is it really 'against any other' or 'against all other'?

l. 378-382: Truly an interesting characteristic if it can be clearly attributed to the imprint of the diurnal cycle. However, do you have more supporting evidence for this then 'just' the line plots? For intensity, one can expect such a relationship, but why should it be the same fore location errors? It could be pure coincidence. By construction, the CDFE metric combines the effects of (a) the sharpness of the ensemble (i.e., how distant the errors of the members are spread around the median) and (b) how much the location of the entire error CDF deviates from 0. Can you check the actual CDFs to see if there are differences between 0 and 12 UTC for (a) and (b), respectively.

l. 382: 'on' → 'in'

l. 382-385: If Fig. 8c+d are removed, reveise sentences and merge with the text describing Fig. 8a+b.

l. 384: 'general' → 'mean'

l.387-388: Significance is either found or not but can't be 'pronounced' from a statistical perspective. You probably mean that it is found for fewer lead times compared to the location results. However, there are also many significant parts for the intensity results, so maybe drop the statement.

l. 383-389: 'The Middle East …': Maybe you can link to Fig. 3a, which shows that there are no very intense cyclones in this region, which is certainly one reason why the intensity is easier to predict. (Maybe it is worth adding relative frequencies for other percentile ranges as panel(s) to Fig. 3, similar to Fig. 3a.)

l. 392: 'firsts' → 'first', and remove 'once again'.

l. 395: 'or' → 'and'

l. 399: change to plural: 'cyclones'.

l. 403: 'the differene remains' → 'differences remain'

l. 406: 'seasons' → 'season', 'The CDFE score' → 'CDFE scores', 'follows' → 'follow'.

l. 406: If significance lines are plotted when a category is different from all other categories, how can it be that SON and MAM are almost on top of each other and yet be significantly different between 48h-102h? Do you have an explanation for this?

Figure 9: Mention here as well why results for SON and MAM are not shown in panel (a)?

l. 408: Different from each other in what sense/variable? MSLPE is shown in Fig. 9b. Are you referring to intensity itself?

l. 411 & 429: Keep using 'categories' as before and avoid 'classes' to be consistent.

l. 412: Change 'Fig. 10a' to 'Fig. 10' as it applies to all panels. Add 'Fig. 10a' after 'TTE' in line 413.

Figure 10: Remove the word 'pressure'.

l. 414: You make a general statement (about independence) but only show results for a single 12h-period (before peak intensity). Have you checked other periods of the cyclone's life time to be so sure?

l. 419: Here you are comparing to papers that 'showed a poor prediction of the intensity', but Fig. 10d presents deepening rates. I suggest to move this sentence to where Fig. 10b is discussed.

l. 421: 'day 4.5' → '108 h'. Also, where is it shown in the paper that 'in our data set is slightly over-predicted from day 4.5 onward'? Are you referring to Fig. 5b? However, the plots exhibits an over-prediction on average even before 108h.

l. 421-424: Maybe is it not just a matter of sampling cases but also of the different regions that are considered.

l. 425: 'from' → 'at'

l. 427: 'the previous part' → 'Section 5.2'.

Figure 11: Add 'moving cyclones, respectively' after 'fastest'.

l. 430: 'based on' → 'in'

l. 431: 'spatial distributions' → 'patterns of relative frequency'

l. 432: Remove 'examined in Section 3.4' as the categorisation here is applied to a different (namely the reforecast) data set. When removed, also change 'for the' → 'for a'.

l. 434: 'from the beginning of the forecast' is only true for the fast category.

l. 445: 'constant' → 'fixed'

l. 447: Inconsistent years when compared to line 95.

l. 450: 'Comparatively': Do you mean 'In comparison' or 'in contrast'? If the former, the rest of the sentence needs a comparative form (e.g. 'higher').

l. 456: 'good reliabbility of the' → 'well calibrated'

l. 457: Add 'on average' after 'observed'.

l. 458: Depending on whether you have followed the suggestion above, change 'for most of the cases' to 'in most forecasts'. Also, add 'median and mean' before 'errors'.

l. 460: In my view, the newly developed CDFE score is not a generalization of the CRPS but a different application of the same concept, namely applied to forecast errors instead of actual forecast values. Also, remove 'probablistic score' as it is part of 'CRPS' already, and replace 'case' by 'cyclone forecasts'.

l. 462-464: Here and in lines 310-311: Why did you split the error growth rate at these lead times? Is there a particular reason?

l. 462: Add 'nearly' or 'almost' before 'constant'.

l. 464: Add 'with lead time' after 'errors', and 'progressive' before 'saturation'.

l. 470: 'the key' → 'a key'

l. 471: Remove second comma.

l. 472: 'important by causing' → 'considerable, as they can cause'.

l. 480: As before, your study does not distinguish between different stages of cyclone life time, nor cyclogenesis nor cyclolysis. Therefore, replace 'cyclogeneses' by 'cyclones'.

l. 484: Add a comma after 'e.g.'.

l. 488: Very minor, but why do you reverse the order of the categorisations applied in your study?

l. 491: 'firsts' → 'first'

l. 491-492: Move 'at high lead times (several days)' to after 'cyclones', and add 'part of' after 'responsible to'.

l. 500: Add '.'

l. 501: 'Is' → 'It'

---

## Author Comment (AC1)

**Response to Reviewer #1**

We thank the reviewers for their time and constructive comments. We have complied with most of the proposed changes. In the following, the comments made by the reviewer appear in black, while our replies are in blue and the changes in the text are in quotation marks.

**General:**

In this paper, the authors examine the predictability of Mediterranean cyclones by using a large reference dataset of 2853 cyclone tracks from 21 years and ensemble reforecasts for this period. This topic is very important for both research and operational scientific communities. The work is ambitious in the way that both location and intensity errors of the cyclone tracks are systematically examined, with decomposition into sub categories of cyclones according to their season of occurrence, sub-region of the Mediterranean, intensity, propagation velocity, and deepening rate. The authors find significant predictability differences among the subcategories in terms of cyclone location and intensity at different lead times. Overall, the paper is very well-structured and clearly written, with good visualizations of the data.

The manuscript will clearly provide a valuable contribution to WCD once the following concerns will be addressed. Of the major comments, the first concerns the underlying tracking methods, while the other two raise issues which are largely not addressed in the current paper, but are nonetheless key for making insightful conclusions (and are feasible given the existing datasets).

**Major comments:**

1. Differences in the cyclone tracking algorithm between the reference and ensemble forecast datasets:

Using non-identical tracking methods for the reference tracks and the forecasted tracks can introduce biases into the analysis. This issue should be either corrected or the differences in tracks carefully assessed for any systematic influence on the predictability metrics.

In this paper, two different tracking methods have been used to track the cyclones in two different data set. The VDG algorithm requires a reference trajectory to be given as input to begin the tracking process. Consequently, and unlike AYRAULT, the VDG algorithm cannot be used to track cyclones directly in ERA5 data without any given starting point. This characteristic makes it however very relevant to track cyclones in the reforecast fields using the reference tracks, without requiring an additional matching method that would bring more complexity. To assess the consistency of the results, the two algorithms have been tested on the same ERA5 data, given the tracks detected by AYRAULT as a reference for the VDG algorithm. Note that VDG is tuned for a temporal resolution of 6 h, while the tracks are followed hourly with the AYRAULT algorithm. For each track detected by AYRAULT and VDG, the difference in terms of location and intensity are calculated for all time-matching track points. The results are presented in Fig. R1-1a and Fig. R1-1b. For 85% of the tracks, no difference between the two techniques is observable. However, for 10% of the cases, the distance between tracks reaches almost 200 km at the time of minimum MSLP. To overcome this issue, tracks are removed from the analysis if: (1) The track is followed with AYRAULT but not with VDG in ERA5 (206 tracks). (2) If the maximal distance between two tracks reaches more than 40 km (687 tracks). With these criteria no difference is observable between AYRAULT and VDG in 99% of the dataset at any instant, for the location or for the MSLP (Fig. R1-1c and R1-1d).
All figures and results have been recalculated using this new dataset of 1960 tracks. The same conclusions are drawn, and the main results remain consistent with the previous version. All values presented in the final text are also modified according to the new robust results. A dedicated part on the verification method has been added (now Section 2.5).

2. Prediction of cyclones per se:

In addition to the systematic investigation of the predictability of location and intensity, it is even more fundamental to understand if the mere existence of the cyclone (or track) is captured by the ensemble. By construction of the verification technique against a reference track, this aspect required a separate quantification,

such as tracking the mean number of members having a cyclone at all (orange line in Fig. 5). In my view, if this aspect is developed, it could greatly enhance the take-home messages from the paper. For example, one can readily decompose this metric (fraction of ensemble members with a cyclone) to the different cyclone categories.

Thank you for your suggestion. A dedicated section has been added in the article (now Section 5.1).

3. Consideration of the cyclone lifetime relative to cyclogenesis/peak intensity:

Currently, the examination of the cyclone tracks predictability does not consider one of the most important aspects, which is the cyclone stage in the lifecycle, namely relative to cyclogenesis/peak time/lysis. Instead, the large variability in Fig. 5 envelops both the variability among ensemble members (for a given cyclone track point), and different cyclones, with all subcategories and at all track times (i.e., the time relative to genesis/peak etc.). It can be insightful to look into this variability, and see how Figs. 5 and 6 differ when decomposing the analysis to different track times. I understand that, by construction, the genesis point is more predictable when forced to start from a vicinity of the reference, but I hope the authors can think of a way to still address this aspect.

The choice to keep all track times has been made to ensure the categories to be sufficiently populated for robust results: unfortunately, the low frequency of reforecast basetimes (twice a week) does not allow decomposing the results in the different cyclone stages. We keep the suggestion for future work, depending on the possible availability of reforecasts with more frequent basetimes.

**Minor comments:**

Line 1 and 20: use of "beneficial" is unclear here

Replaced by "necessary"

Line 4: "characterize cyclone predictability": mention the timescale / range you consider here

Correction: "medium-range" added here

Line 22: add "negative" before "impacts"

Done

Paragraph ending in line 31: the paragraph currently misses a description of heat lows over land.

Added

Line 47: mention that Baumgart et al. considered hemispheric-wide simulations

Correction: "Earlier work by "Zhang2007" in an idealized baroclinic wave simulation, and by "Baumgart2019" in hemispheric-wide simulations of PV structures, identified three phases in forecast error growth"

Line 108: 1) the usage of data on the 700 hPa level comes as a surprise and was not mentioned before. Please clarify. 2) Also on line 111: is the smoothed relative vorticity field evaluated at both 850 and 700 hPa? 3) Line 120: "...maximum remains the track point..." - on which level?

1) Correction: "The main idea of AYRAULT is to track cyclones firstly in the relative vorticity field at 850 hPa. The horizontal wind is then used at both 700 hPa and 850 hPa to choose the best following tracking point in the direction of cyclone propagation. Finally, the track points are paired with the mean sea level pressure (MSLP) field. In the following, a time step is denoted by $t$, the relative vorticity field at 850 hPa by $\zeta$, the zonal and meridional wind fields by $u$ and $v$, respectively". 2) and 3) The relative vorticity is explicitly defined as to be at 850 hPa only: "the relative vorticity field at 850 hPa by $\zeta$"

Lines 116-117: how is the selection made? Do you mean that closer and more intense is favoured?

Clarified: "the strongest one"

Lines 130-131: it is unclear then why not to use a lower confidence interval to obtain a comparable number of tracks. Please also mention how many tracks are obtained here?

Correction: "First, the latter contains only 206 tracks in the highest confidence level, which is not enough for a systematic study. At the mean confidence level (consensus of 5 over 10 algorithms), on the 2001-2021 period and with the same thresholds used here on the pressure and on the location of cyclones, 1231 tracks are detected in "Flaounas2023", against 2853 with the AYRAULT algorithm."

Line 164: "combination" = mean?

Yes. Correction: "The tracking method in VDG is then based on a combination of past movement and steering flow vector $V_{av}$ defined as the layer average of the local wind fields at 850 hPa and 700 hPa."

Fig. 3a: it is unclear how the subset density is normalized (% of the strongest 10% of all cyclones?), same for Fig. 4.

Yes, it is in % of the strongest 10% of all cyclones. Correction: "(a) Relative frequency of Mediterranean cyclones, defined as the 10 % deepest cyclones having a track point within a radius of 100 km."

Lines 305-306: please clarify if the "number of members in which a cyclone is found" means that there is a cyclone track point within a 3.5-degree radius near the reference track?

No, when the VDG algorithm found a track point at time $t$ corresponding to the reference track, it can move independently forward at time $t + 1$. Added in the description of the VDG algorithm: "At initialisation time, a $\zeta$ maximum is searched for in the reforecast field in the neighbourhood of the reference track calculated in ERA5. The tracking in VDG is then independent of the reference track, and is based on a combination of past movement and steering flow vector..."

Fig. 5: Bringing the orange line to the front can make it more readable

True, but the highlight is put here on the distribution, not on the precise number of members. Moreover, this curve is now presented in the next section.

Line 377: good prediction of W. Med cyclones: it can be expected because they are mostly slow-moving, so errors in location are not expected to arise compared to faster moving systems.

Yes, detailed later in the section on the motion speed categories.

Line 389: add "36 h," before 42.

Modified by the new results: "66 h, 90 h, 108 h and 132 h lead time,"

Fig. 10c,d and accompanying text: this view is confusing, as it does necessarily capture the cyclone at the timing of the maximum deepening (12 h until max. intensity). It will be more interesting to isolate only this section of the track and examine its predictability metrics.

See response to major comment 3

Line 428: Need to reconcile this statement with the best predictability of W. Med cyclones in terms of track location. It will be good to come back to this issue in the conclusions section.

The issue is addressed in the motion speed part. "Indeed, these quasi-stationary lows are in a vast majority concentrated in the Gulf of Genoa in the West Mediterranean, which is the region where the cyclone location is the best predicted (see Fig. 9). This result has to be compared with the poor predictability of the West Mediterranean cyclones' intensity, suggesting the existence of at least two different types of cyclones in this

particular region. The first is made of slow cyclones with a good predictability in terms of location and a fair predictability in terms of intensity. The second one is made of fast moving cyclones with a poor predictability in terms of intensity and a fair predictability in terms of location. "

Line 480: cyclogenesis: this stage is no directly shown here (see also major comment 3).

Replaced by: "Winter cyclones are indeed less accurately predicted"

**Technical corrections:** Done

Line 54: robuster => more robust

Line 70: systematical => systematic

Line 107: in => into

Lines 192/384/392: delete "for its/their part" without loss of information

Line 209: spell out "CDFE"

Line 213: add "." after "applied"

Line 247-248: "do not enter over the sea" - unclear wording

Line 306: on => in

Line 315: of => to

Fig. 5 caption: in function => as a function

Line 382: "the both" - delete "the"

Line 392: firsts => first

Line 399: add "visualized" after "previously"

Line 406: every => all

Line 436: statically => statistically

Line 473: a same area => the same area

Line 491: firsts => first

**Additional figures**

[Figure]

Figure R1-1: Comparison between Ayrault and VDG tracks obtained from ERA5: distance (a and c) and absolute MSLP difference (b and d). (a) and (b) show the differences in the original datasets, while (c) and (d) show the differences after removing tracks with a distance reaching more than 40 km (final data sets). 0 h lead time corresponds to the time of minimum MSLP in VDG.

---

## Author Comment (AC2)

**Response to Reviewer #2**

We thank the reviewers for their time and constructive comments. We have complied with most of the proposed changes. In the following, the comments made by the reviewer appear in black, while our replies are in blue and the changes in the text are in quotation marks.

The aim of this paper is to investigate the predictability of Mediterranean cyclones when conditioning on different categories, namely region, season, cyclone intensity and motion speed. In a first step, reference tracks are identified in ECMWF's ERA5 data set and results are presented in section 3, documenting the climatology of that data set and discussing commonalities with and differences to previous studies. The reference cyclones are then tracked in ECMWF's 11-member ensemble reforecasts to compute location and intensity errors as well as different measures for means, median and spread. These are discussed together with an assessment of forecast reliability in section 4. Finally, practical predictability is evaluated condition on the categories mentioned before, using a novel approach that defines CDFE scores. The underlying idea stems from the formulation of the CRPS and integrates the area between an error CDF and the Heaviside function with the step at 0. The results show that the predictability of cyclone location is mainly determined by cyclone motion speed, while the intensity and deepening rate are found as the two key factors that determine how good the intensity is predicted. These results are an important first step to identifying and better understanding the main drivers for predictability of Mediterranean cyclones. However, I strongly suggest to rephrase the title as there are many more factors that determine predictability than the four aspects studied here. The paper is well structured and written and the results are generally clear. However, there are two main concerns: First, the cyclones comprise a broad range of storm natures (extra-tropical to tropical-like), which may have different predictability associated with different natures, but this is not addressed at all. Second, two different tracking algorithms are applied to the reference and reforecast data, which may unnecessarily affect the calculation of errors, and hence the results of the predictability study.

I recommend the paper for publication after consideration of the general comments below. Specific comments are also provided to help improve the paper.

**General comments**

1. Title: I find the words 'What determines' too open for what is covered in the paper. The study looks at differences in predictability based on the region, season, intensity and motion speed, respectively. And these factors are definitely important. However, the word 'what' could also refer to differences in precursors, the influence of the environment or physical processes (as mentioned in the outlook paragraph in the summary and conclusions section). I therefore suggest specifying the title so that it better reflects what is being investigated.

The title was changed to "Systematic evaluation of the predictability of different Mediterranean cyclone categories" to better reflect the contents of the paper.

2. As described in the introduction section, there is a certain range in the nature of cyclones in the Mediterranean region, spanning from deep extratropical cyclones to tropical-like Medicanes. Therefore, it can be assumed that the predictability also depends on the nature of the storm, as different processes may play a role during formation. However, the paper makes no distinction on this point, which is why it is questionable to what extent the results are comparable to other regions (the paper often compares with the Atlantic or the whole northern hemisphere), as they may cover a different range on the spectrum of storm natures with different frequency distributions. If examining this aspect in the reforecasts is considered to be beyond the scope, I yet recommend to add some analysis and discussion in this regard for the reference tracks, to at least document the distribution of storm natures for your data set. There are different approaches to this, but one way would be to calculate CPS metrics, for which code would be available, for example here: `https://github.com/fredericferry/era5_cps_diagram`

As stated in the introduction, medicanes received interest of the scientific community in recent years, and these phenomena can produce severe winds and rainfall as in the cases of Ianos in September 2020 or Daniel in September 2023. We agree that medicanes are very interesting cyclones and better understanding their peculiar

dynamics in the tropical-extratropical continuum is a crucial point for the Mediterranean cyclones community. However, as added, "they are very rare with 1 to 2 events per year (Cavicchia et al. 2014)." This number can be compared to the about 100 cyclones per year on average in our final dataset. While medicanes tend to be over-represented in the current literature, we follow a more systematic approach here including the 99% of cyclones that are not tropical-like. It was clarified that "the statistical signal of medicanes can be considered as negligible".

3. Apart from this additional analysis, the term 'Mediterranean cyclone' should be defined more precisely in terms of what range of cyclone natures is spanned. Also, please check the entire manuscript to see where the term 'cyclone' can be specified more precisely in order to improve clarity on this point.

Now stated in the introduction: "Therefore, the present study mainly focuses on the predictability of extratropical cyclones in the Mediterranean". The extratropical nature is now also specified wherever relevant in the manuscript.

4. You introduce the two 'types' of predictability (practical vs intrinsic) in the introduction, but never use these terms thereafter again. Since your study is based on forecasts from the IFS model, I suggest to clearly state once that the predictability you are assessing is of the type 'practical', and then say: 'hereafter referred to as 'predictability' for simplicity'.

Now stated in the introduction: "In the following study, the 'practical predictability' will be denoted by 'predictability' for simplicity."

5. Cyclone detection in the Ayrault algorithm: According to lines 119-120, the resulting track may be a mixture of $\zeta$-based and MSLP-based points. Why are tracks created based on such an inconsistent use of variables? Since the study uses error distributions of cyclone location (and intensity) as the basis to investigate predictability, I consider this point to be crucial with regard to all conclusions based on it. Could you check how different reference tracks are, when solely using $\zeta$-based track points?

After a careful verification, we found that the algorithm finds a MSLP track point associated with every $\zeta$ maximum, thus all tracks are consistent. It has been corrected in the text.

6. Why is a different tracking method used for the reforecasts? For me, it would be more obvious to use the same method for both data sets. In lines 158-160, you say that you refrained from applying the Ayrault algorithm to the reforecasts. But why did you not apply the VDG algorithm to both datasets (i.e., reforecasts and ERA5), to keep it consistent? Is there any particular reason? One downside of this inconsistency is that the 'data and methods' section becomes (in my opinion unnecessarily) longer.

In this paper, two different tracking methods have been used to track the cyclones in two different data set. The VDG algorithm requires a reference trajectory to be given as input to begin the tracking process. Consequently, and unlike AYRAULT, the VDG algorithm cannot be used to track cyclones directly in ERA5 data without any given starting point. This characteristic makes it however very relevant to track cyclones in the reforecast fields using the reference tracks, without requiring an additional matching method that would bring more complexity. To assess the consistency of the results, the two algorithms have been tested on the same ERA5 data, given the tracks detected by AYRAULT as a reference for the VDG algorithm. Note that VDG is tuned for a temporal resolution of 6 h, while the tracks are followed hourly with the AYRAULT algorithm. For each track detected by AYRAULT and VDG, the difference in terms of location and intensity are calculated for all time-matching track points. The results are presented in Fig. R2-5a and Fig. R2-5b. For 85% of the tracks, no difference between the two techniques is observable. However, for 10% of the cases, the distance between tracks reaches almost 200 km at the time of minimum MSLP. To overcome this issue, tracks are removed from the analysis if: (1) The track is followed with AYRAULT but not with VDG in ERA5 (206 tracks). (2) If the maximal distance between two tracks reaches more than 40 km (687 tracks). With these criteria no difference is observable between AYRAULT and VDG in 99% of the dataset at any instant, for the location or for the MSLP (Fig. R2-5c and R2-5d).
All figures and results have been recalculated using this new dataset of 1960 tracks. The same conclusions are drawn, and the main results remain consistent with the previous version. All values presented in the final text are

also modified according to the new robust results. A dedicated part on the verification method has been added (now Section 2.5).

7. As the word "velocity" can be used and understood in different contexts, I suggest using 'motion speed' throughout instead.

"Velocity" was reworded to "motion speed" as suggested.

8. I suggest to harmonise the titles of sections 5.1-5.4. Either use 'Differences in X' or 'X categories'.

The titles have been harmonized as suggested.

**Specific comments**

l. 1: 'in a densely populated area' is very unspecific. Rephrase so that the geographical location does not have to be implicitly deduced from the term "Mediterranean cyclones".

Correction: "Cyclones are essential components of the weather patterns of the densely populated Mediterranean region, providing necessary rainfall for both the environment and human activities."

l. 5–6: 'this region': Make sure to have clearly stated, either here or in the first sentence, which region you are referring to.

Correction: "The aim of this work is to characterise the cyclone predictability in the Mediterranean"

l. 5–6: Incorrect name: ECMWF is the "European Centre for Medium-Range Weather Forecasts".

Corrected

l. 16-17: "opens the way to identify the key processes..." is what a reader may expect when reading the title, but it is not addressed in the present manuscript. Therefore, I strongly recommend to specify the title as mentioned in one of the general comments.

See response to major comment 1

l. 19: The first sentence of the introduction is very similar to the first sentence of the abstract. However, while Mediterranean cyclones are described as a 'component of the climate' in the abstract, extratropical cyclones are introduced as 'component of weather patterns' in the former. Is there a particular reason why they are introduced in conjunction with different temporal (and spatial) scales? If so, please clarify, otherwise I would recommend to use consistent wording.

Made consistent: "of weather patterns in the mid-latitudes."

l. 24: Replace 'life cycle' by 'lifetime'.

Corrected

l.24: The previous paragraph specifically addresses extratropical cyclones. Here you make a comparison between cyclones in the Mediterranean and other basins, not mentioning the storm's nature. For the statements you make, are you comparing extratropical / subtropical / hybrid / tropical(-like) cyclones in the Mediterranean and other basins? Please clarify.

Correction: "In the Mediterranean, extratropical cyclones are generally smaller and with shorter lifetime than in other larger basins". See also response to major comments 2–3.

l. 24-36: Only with the introduction of medicanes in l. 36 you begin to discuss cyclone nature for the Mediterranean. Please make sure that a reader knows which type (or range on the spectrum) you are addressing

in these lines.

Clarified. See also response to major comments 2–3.

l. 47: Note that Baumgart et al. (2019) did not investigate error growth in the specific context of cyclogenesis and cyclone prediction, respectively. Consider this in the text.

Replaced by: "Earlier work by "Zhang2007" in an idealized baroclinic wave simulation, and by "Baumgart2019" in hemispheric-wide simulations, identified three phases in forecast error growth."

l. 53: Replace 'hardly' by 'rarely'

We prefer keeping the original wording.

l. 54: The comparison seems to be inappropriate. The purpose of the ensemble is to quantify the uncertainty beyond a purely deterministic prediction, and hence provides an additional perspective. The word 'robuster' (should be 'more robust') only makes sense if one compares a single forecast with an ensemble mean.

Robustness may actually depend on the decision-making process (see `https://agupubs.onlinelibrary.wiley.com/doi/full/10.1002/2017EF000649`). Here we use the word as in "perform well under a wide range of plausible conditions" (which implies extreme events in that specific sentence). It is possible that the Reviewer has another definition of robustness in mind that would only apply to single forecasts. But, in our current understanding, we maintain that the full ensemble gives a more robust forecast than a deterministic forecast. Indeed, extreme events are more likely to be predicted in at least one of the member of an ensemble than with a single forecast. Moreover, when two (or more) distinct scenarios are emerging, the ensemble mean would give some middle ground in between the scenarios with a much lower probability to happen. Therefore, we think it is legitimate to say that ensemble forecasts perform better than deterministic forecasts under a wide range of conditions, which is phrased in our sentence as "providing more robust results". (Correction noted for "more robust".)

l. 56: There are more intensity categories below the hurricane intensity threshold. Therefore, bringing together the word 'hurricane' and 'genesis' seems a bit odd. I suggest to say 'tropical cyclone genesis' instead.

Corrected

l. 64: Add comma before 'but'.

Done

Also, I suggest to think of (and write about) Medicanes as a subset of the 'Mediterranean cyclones', instead of emphasising the lack of representativeness.

See response to Major Comments 2–3

l. 66: What nature (or spectrum of cyclones) of cyclones are you referring to?

Clarified: "Noteworthy, "Picornell2011" assessed the deterministic forecast quality for more than 1000 extratropical cyclones during a whole year"

l. 69: Replace 'firsts' by 'first'.

Done

l. 70: Replace 'systematical' by 'systematic', and remove 'as objects'.

Done

l. 71: What type of vorticity? Relative vorticity?

Correction: "They tracked the cyclones in global forecast data for two winter and two summer periods and defined errors in both location and intensity based on maximum relative vorticity compared to analysis data"

l. 72: Replace 'degree by 'degrees'.

Done

l. 74: Replace 'and this' by ', which'.

Done

l. 76: Since this paragraph reports on results from previous studies, mainly addressing model deficiencies ('incorrect representation','systematic slow bias', etc.), it's probably worth specifying the predictability here and in l. 69 as 'practical predictability'. But I leave it to the authors to decide.

See response to major comment 4

l. 79: What is meant by 'with a homogeneous configuration'. Please clarify.

Correction: "The forecast model covers a 20-year period with the same configuration, which allows extracting robust statistical signals."

l. 79-80: Replace 'robust statistical signals' by 'statistically robust signals'.

Done

l.81: In the main text, you have not referred to specific ensemble prediction system yet, so either remove 'the' before 'ensemble' or state which one you are going to use.

Corrected: "Their representation in an ensemble prediction system is discussed"

l. 84: Add 'cyclone' before 'tracking'.

Done

l. 91: Add a hyphen between 'Medium' and 'Range'.

Done

l. 91-92: Remove 'hereafter'.

Done

l. 95: If you have used a regular lat-lon grid, replace 'with 0.25° horizontal resolution' by 'on a 0.25°x0.25° horizontal grid', as it is relevant technical information (for the purpose of reproducing the method/results).

Done

l. 98: Replace 'need' by 'step'.

Done

l. 100: 'Coarser' than what? Can you give an approximate number?

The original algorithm was run using ERA-15 reanalysis with approximately 125-km horizontal resolution. However, the algorithm has evolved since then and been applied to intermediate resolution data. We prefer not

discussing this historical evolution and simply replace "coarser" with "coarse".

l. 101: Replace 'required a specific tuning' by 'had to be adapted', 'Indeed' by 'As stated before, 'with' by 'have' and 'life cycles' by 'lifetime'. Also, add 'those' before 'in the'.

Done

l. 102: Add a comma before 'and'.

Done

l. 103: Add 're-' before tuned.

Done

l. 104: Add 'to' before 'operate'.

Sentence changed: "The main idea of "Ayrault98" is to track cyclones firstly in the relative vorticity field at 850 hPa. The horizontal wind is then used at both 700 hPa and 850 hPa to choose the best following tracking point in the direction of cyclone propagation. Finally, the track points are paired with the mean sea level pressure (MSLP) field."

l. 107: 'in' à 'into'.

Done

l. 108: Is there any reason why you have chosen these characteristic lengths? If so, please add a sentence stating it. Maybe also consider adding a sentence explaining the reason for the smoothing (i.e., why you apply a moving average). And since you consider a broad range of cyclones (large extratropical to smaller tropical-like), did you test whether/how your choice works equally well when tracking different cyclone types?

All the tunable parameters mentioned have been tuned specifically to provide the best tracking performance in the Mediterranean, including the length of smoothing of 225 km. The starting point for the tuning was the parameters from `https://doi.org/10.1007/s00382-016-3394-y` (now mentioned in the text), which has then been adapted to the ERA5 dataset. In particular, 225 km smoothing applied on vorticity is the best compromise for keeping a sufficient number of relevant vorticity cores: lower smoothing length would keep too many cores, and higher smoothing length would destroy relevant ones. 280 km smoothing on wind is to keep the environmental direction and avoid including the wind associated to the vortex.

l. 111: To avoid confusion with the two levels used for winds, add 'at 850 hPa' after 'smoothed field'.

Done

l. 112: When retaining the strongest maxima, is this evaluated for each grid point separately on the full set of maxima identified with then 300 km radius or does the dropping of maxima affect the search at grid points in the neighbourhood? Depending on the procedure, revise the description accordingly to ease reproductibility.

The right answer is the first option (this is evaluated for each gridpoint separately). For reproducibility, look at find_allmin() of `https://github.com/UMR-CNRM/Traject/blob/main/src/Tools.py`

l. 113: 'on' by 'over', and add 'in a given reforecast' after 'time steps'.

"over" added. "in a given reforecast" would be wrong here, because the Ayrault algorthm is applied here to tracking cyclones in ERA5 only

l. 114: Replace 'three steps' by 'three-step'.

Done

l. 115: Add 'for time t+1' after positions, and remove the word 'for'.

Done

l. 115: How large is the neighbourhood? Please state it clearly in the text.

Done. Only one $\zeta$ maximum remain every 150 km (radius of 300 km)

l. 116: The term 'by taking into account' is not a precise methodological description that would allow for reproducibility. Please clarify how that distance value variation (which I assume is variance?) is used in the quality criterion, to allow the reader to understand how the final position is determined.

Reproducibility is ensured by making the code open-source and documented. The algorithm has many technical details that we choose not to put in the text for the sake of length and alleged interest of the readers of WCD. We would rather refer the readers interested in such details to the documentation of the algorithm: https://github.com/UMR-CNRM/Traject/

l. 119: Replace 'in' by 'within', and add a hyphen between '3°' and 'square'.

Done

l. 121: Replace the second 'of' by 'if'.

Done

l. 121-124: In line 120 you describe that it can happen that for some $\zeta$-based track points there are no matching MSLP points identified. In such cases, 'the $\zeta$ maximum remains the track point'. How can you check validation criteria based on MSLP, when there are track points, for which no match was found? What MSLP value is associated with such points instead?

After double-checking, the algorithm finds in every case a local MSLP minimum. Correction: "Pairing with MSLP: for every point belonging to the track, the local minimum of MSLP located within a 3°-square centred on the $\zeta$ maximum becomes the new track point."

l. 123: Replace 'life cycle' by 'lifetime'.

Done

l. 124: Swap words: 'secondary local' à 'local secondary', and replace 'northern' by 'Northern'.

Done

l. 125: Replace 'that enter into' by 'entering'.

Done

l. 126: By 'areas', are you referring to the areas that will be defined later? If so, it can't be used here because it has not been introduced yet. Maybe remove the word.

No, we refer to the areas covered by water of the Mediterranean and of the Black Sea. Correction:"Finally, an additional criterion is applied to only retain tracks entering into either the Mediterranean Sea or the Black Sea".

l. 127: Is 'the Mediterranean-adapted version of the Ayrault (1998) algorithm' referring to the algorithm you are using in your work or another algorithm? To me, it sounds like it describes the adaptations you did to the Ayrault (1998) algorithm. Please clarify. And replace 'slight' by 'slightly'.

Referring to the one used in the study. Correction: "The Mediterranean-adapted version of the "Ayrault98" algorithm previously described has been successfully tested with a slightly different configuration"

l. 129: 'distribution' à 'distributions'

Done

l. 131: What does 'in the highest confidence level' mean? Please clarify.

Correction: "First, the latter contains only 206 tracks in the highest confidence level (*i.e.* consensus of the 10 algorithms), which is not enough for a systematic study."

l. 134: Since the reference tracks are also used as part of the predictability study, I would replace 'predictability study' by 'reforecast tracks', to be consistent with the title of subsection 2.1.

Correction: "Data for the reforecast tracks: IFS ensemble"

l. 135: There is no need for a period per se, you could just run a single reforecast. Therefore, I suggest to replace 'on a historical period spanning typically several decades' by 'starting from historical initial conditions'.

Done

l. 136-137: Given that you already state in the previous sentence, that 'reforecasts are forecasts ...', I don't see the need for this sentence and would hence remove it. Together with the previous comment, it should be clear that they are run from past initial conditions (without taking into account observations afterwards). If you remove, also remove 'therefore' in line 137.

Removed

l. 146-147: 'life cycle' à 'lifetime'

Done

l. 147: Remove 'for'.

Done

l. 150: Replace 'of' by 'for'.

Done

l. 150-151: Having read the last two sentences of this paragraph several times, I am not sure I'm getting your point. Please revise it to make it clearer.

Corrected: "Note that the same cyclone can be tracked in two successive forecast initialisations. When this happens, the two forecasts are treated independently."

l. 152: Same as for the title of section 2.3: 'predictability study' to 'reforecast tracks'

Done

l. 154: Given that you applied a different tracking method to the reforecasts, it is worth mentioning why already after the first sentence of this paragraph? I suggest to move lines 158-160 after '... tracking of tropical cyclones.' (in line 154).

We prefer to let it here, as the explanation of the "why" is detailed afterward

l. 158: Replace 'track predicted cyclones' by 'tracks predicted for cyclones'.

Correction: "This characteristic is particularly useful when it comes to track cyclones that were previously identified in the reference data set."

l. 159: To make clearer that the 'additional step for matching the cyclones' is needed for cyclones (first identified at lead times 0) and becomes both more important and challenging as lead time increases, I would replace 'the cyclones' by 'forecasted and observed cyclones, which are first identified at lead times greater 0'. I would remove 'subjectivity' as there are several of objective methods in the literature addressing exactly this issue.

Correction: "Applying "Ayrault98" to the reforecasts would have required an additional step for matching forecasted and observed cyclones, bringing more complexity."

l. 161: Add 'time' after 'initialisation'. Also, here and where necessary throughout the paper, replace '$\zeta$ maximum' by '$\zeta$-maximum'.

Done

l.162-163: The word 'combination' is used twice, replace ones.

Correction: "The tracking in VDG is then independent of the reference track, and is based on a combination of past movement and steering flow vector $V_{av}$ defined as the layer average of the local wind fields at 850 hPa and 700 hPa."

l. 163: 'position' to 'positions'.

Done

l. 167: Replace 'and here set at 0.5, and $\delta$t the time step' by 'of the forecast $\delta$t, but here set to 0.5'.

Done

l. 170: Replace 'using' by 'within' and 'around' by 'on'.

Done

l. 175: Add the statement that you did not apply the other two requirements as in the Ayrault algorithm (i.e., a minimum track length of at least 24h and a minimum intensity of 1005 hPa that has to be reached), and maybe say why. Probably, because you know that there was a cyclone observed based on ERA5 but you don't want to restrict your ensemble reforecasts too much. Correct?

Correct. Added: "The validation criteria assuring that cyclones last longer than 24 h and reach at least 1005 hPa along their lifetime, which were applied with the "Ayrault98" algorithm to construct the reference data set, are not applied here in the reforecasts."

l. 176-177: A very important information that I would suggest to tell the reader before the VDG algorithm is described in detail! I'm sure it will help avoid some questions that may arise.

Moved before the VDG description

l. 181-182: remove 'with each other, whether in location or in intensity', as it is a general condition.

Done

l. 183: 'for each cyclone track' is ambiguous as there are observed and (for a unique cyclone usually multiple consecutive) forecast tracks. I suggest to remove it and replace 'defined' by 'calculated'. Also add 'each' after 'of' and remove 's' from 'members'.

Done

l. 184: 'the cyclone life cycle' to 'a cyclone's lifetime'.

Done

l. 184-185: Is there a particular reason why you chose to calculate spread as the mean of pairwise differences rather than the standard approach based on the ensemble mean? (By ensemble mean one would of course consider only the subset of members that have a cyclone at time t.)

Yes. Only the spread is defined by the pairwise difference between members. Errors are calculated independently for each member to avoid the need of averaging 10 values into 1, which is an unnecessary summary of the signal in our case.

l. 188-189: Remove the four words 'respectively'.

Done

l. 190: 'defined' to 'calculated'

Done

l. 191: remove 'an'. Also add 'forecast' after 'for each'. 'life cycle' to 'lifetime'.

Done

l. 195-196: Remove the two words 'respectively'.

Done

l. 204: remove 'resp.'.

Done

l. 211: Replace 'CDFE($\tau$)' by 'CDFE($F_\tau$)'.

Done

l. 213: Add '.' after 'applied'.

Done

l. 213-214: Replace 'A higher CDFE (respectively the smaller) indicates a poorer (respectively a better)' by 'A higher (smaller) CDFE indicates a poorer (better)'.

Done

l. 215: With your significance test, You can only test for either similarity or difference at a time, as you would need to define a different null hypothesis otherwise. Remove 'respectively', and add 's' after 'CDF' (plural).

Done

l. 215: Replace '(respectively different)' by 'or not'.

Done

l. 221: Add 'the' before 'toponyms'. Figure 1: The spatial distribution is only one aspect. Isn't it the (relative?) frequency of occurrence what you are showing? This also includes the temporal aspect. Also, remove 'the' in the last sentence. For (b), add a note that the informs about the nonlinear shading levels.

Correction: "(a) Elevation map over the Mediterranean domain with the toponyms mentioned in the text. (b) Relative frequency of Mediterranean cyclones, based on ERA5 over the 2001-2021 period, defined as the percentage of cyclones having a track point within a radius of 100 km. Regions of interest are framed by black boxes. Note that the shading scale is not linear."

l. 223-249: Following the previous comment, I would suggest to replace 'spatial distribution' by 'relative frequency'. If you change it, make sure to also change it in line 220.

Done, but not changed in l220, because the part is explicitly dealing with the spatial distribution aspect.

l. 224: Add a comma after 'period'.

Done

l. 228-229: Add commas before designed and after areas.

Done

l. 231: The reference (Trigo et al. 2002) for this statement made here is only mentioned a couple of lines later. Can you include it here already?

Done

l. 247: 'at' à 'of'

Done

l. 248-249: Since it is not found in most of the other studies, do you have an idea of potential (maybe methodological?) reasons for this? If so, it would be worth to mention.

No, but we believe the result is worth noting.

l. 248-249: Having read lines 125-126 again, I wonder how this condition is actually defined. Does 'enter into' refer to the domain enclosed by the box (as shown in Fig.1b) or to the entire sea basin? If the former, does it matter whether a cyclone enters the box over land vs over sea? Please clarify.

Correction: "entering into either the Mediterranean Sea or the Black Sea."

Figure 2: Replace 'on' by 'over'. What does 'in their mature phase' mean? Is Fig. 2 only based on the most intense fragments of your reference tracks? If so, it should be stated clearly how this is filtered.

Correction: "Monthly number of cyclones at their minimum MSLP point in the six regions defined in Fig. (averaged over the 20-year period)."

l. 267: 'rate' à 'rates'

Done

l. 267-268: Can you add a reference for the link to the Atlantic?

Correction: "While cyclones in these two areas are influenced by the Atlantic (Raveh-Rubin2017)"

l. 269-279: Here you discuss the depth of cyclones, but it is never stated in the main text (but only in the caption of Fig. 3) how these categories are defined. Please clarify either here or in the 'data and methods' section.

"The 10% deepest cyclones" was added in the text.

Figure 3: As before, consider replacing 'spatial distribution' by 'Relative frequency'. I suggest to move the

definitions of the categories to the main text and then add something along the lines 'See the main text for details on the category definitions.'. For (a), add a note that the informs about the nonlinear shading levels.

Correction: "(a) Relative frequency of Mediterranean cyclones, defined as the percentage of the 10 % deepest cyclones having a track point within a radius of 100 km. Note that the shading scale is not linear. (b) Monthly mean number of cyclones in the 3 categories of intensity. Each category contains 10 % of the tracks data set, *i.e.*, the 10 % deepest (green curve), the 10 % around the median intensity (orange curve) and the 10 % shallowest cyclones (blue curve)." We prefer to let the technical details in the legend of the figure to keep the scientific purpose clear in the text.

l. 279: As suggested in the general comments, I would replace 'velocity' by 'motion speed', and remove 'of Mediterranean cyclones' to stay consistent with the previous subsection titles.

Done

l. 280: Why have you chosen the median and not the mean? Is it because the distribution is so skewed? Probably worth to mention. How large are the differences to the mean? Also, 'life cycle' to 'lifetime'.

We prefer using the median here for consistency, as it is applied to the median motion speed (see Fig.R2-1). The difference is small actually, with 27.8 km h$^{-1}$ for the mean and 25.3 km h$^{-1}$ for the median. The skewness of the distribution shown in Fig.R2-1 is already suggested in the text: "According to our calculations on the reference data set, Mediterranean cyclones move eastward on average at a median motion speed of 25 km h$^{-1}$. However, the variability is large and the fastest 5 % are moving at velocities greater than twice the median." 'Lifetime' corrected in the text.

l. 281: Replace 'generally move from the west to the east' by 'move eastward on average' as there are cases where it moves westward (e.g. Medicane Daniel). Can you also add a statement on the meridional component?

Done. The meridional component is variable from a region to another, and there is no clear average behaviour (see Fig. R2-2).

l. 282: Remove 'the' before 'twice'.

Done

l. 283 & l. 285: As is section 3.1, replace 'spatial distribution' by 'relative frequency'. Figure 4: As with Fig. 1b+3a, replace 'spatial distribution' by 'relative frequency'. Add a note that the informs about the nonlinear shading levels.

Done

l.284-285: 'change' to 'changes', and replace 'from one velocity-based category to another highlights' by 'between the motion speed-based categories highlight'.

Done

l. 296-297: As you do not distinguish between different stages of a cyclone lifetime in your study, I would refrain from linking different motion speed-categories to differences in processes of cyclogenesis. It may be that a cyclone forms in one part of a region but then moves into another part of the same region, while changing motion speed. Therefore, here and throughout the paper, I recommend to not speculate on cyclogenesis, if it is not specifically analysed.

Indeed, the motion speed of a cyclone may vary during its lifetime. However, the areas highlighted on Figure 4a are clearly distinct from each other and it is very unlikely the same cyclone would move from one to the other. Also, note that we do not explicitly refer to the cyclogenesis area but more broadly to where cyclones arrive from (the Sahara, the Atlantic).

l. 302: I suggest to use a more concise title for 4.1: 'Location and intensity errors'.

Done

l. 303: Remove second 'in', and 'Fig.' à 'Figure'.

Done

l. 304: ', for the all' à ' for the entire'.

Done

l. 305: It is not clear to me which hypothesis you want to test, which is why the topic of significance is not appropriate here. Presumably, you want to say that the large number leads to more stable results!?

Correction: "ensures the results to be more robust".

l. 306: Add 'approximately' before 'linearly', and 'on' to 'in'.

Done

l. 308: 'until' to 'up to'.

Done

l. 309: '3 day' to '72 h', to be consistent with the x-axis.

Done

l. 310: 'not fully linear': Instead, I suggest to describe how it actually is, namely 'slower-than linear'.

Done

l. 311: As you talk about the TTE growth rate x in the previous part of the sentence, 'it increases' is misleading, as the growth rate is actually decreasing. If you replace 'it' by 'TTE', it should be fine.

Correction: "in the first 78 h, the median TTE increases of about 40 km per day, while from 84 h lead time and beyond it increases at a smaller rate of about 18 km per day"

l. 312 & 316: Maybe replace 'processes' by 'reasons'.

Done

l. 314-315: I find it difficult to understand what point you are trying to make here. Please clarify and rephrase the sentence. Also, do not use 'early lead times' and 'from a few hours', as it is not clear what it means. Use 'short(er)' or 'long(er) lead times' instead. And change 'i.e' to 'i.e.,'.

The previous sentence was confusing and has been removed. The sentence now reads "Consequently, when the lead time increases, the proportion of cyclones followed since early lead times (*i.e.*, which forecast track may have diverged from the reference track) decreases in comparison to the ones followed since long lead times (*i.e.*, which forecast track is still in the neighbourhood of the reference track, by construction)."

l. 316: 'deals with' to 'has to do with'.

Done

l. 316-317: In principle, a forecast can predict anything, but what we want is that it ultimately converges to climatology. Since an ensemble forecast is considered here, 'a random climatological state ...' is not adequate

wording. Maybe say: 'For long enough lead times, an ensemble forecast should ultimately converge to the climatological distribution'.

Replaced.

l. 318: Remove 'a' and change 'value' to 'values', as mean and median may not necessarily converge to the same value if the climatological distribution is skewed.

Done

l. 321: What maximal growth rate do you get, when calculating it over the first 48 h only, as Picornell et al. (2011) did?

Maximal growth rate is misleading here. Correction: "Overall, the growth rate of 40 km per day in the first 78 h lead time is remarkably close to the 43 km per day found by Picornell2011 in the Mediterranean".

l. 322: A comparison with the "whole Northern Hemisphere" means comparing to a broader spectrum of cyclone natures (e.g. including true tropical cyclones), and thus different error characteristics (i.e., distributions). If you want to keep this comparison, you should point out this important difference. Given the much coarser resolution of that study, I wonder whether it would be better not comparing to it.

Froude2007 do not consider the tropical band in their study. The important difference is already pointed out. Corrections: "In the extratropical Northern Hemisphere, and using the operational ensemble prediction system of the ECMWF from January to July 2005, Froude2007b found a much higher mean error growth rate of 1.25° (about 137 km) per day and almost constant until 7 days lead time. The coarser resolution of the ensemble prediction system used in their study (about 80 km) and the particular characteristics of Mediterranean cyclones could explain this difference in the mean error growth rate."

l. 327: Remove 'In our case,'.

Done

l. 328: 'from' to 'at'. Also, replace 'are slightly late' by 'have slow propagation speed bias'.

Done

l. 329: 'in average' à 'on average'. Also, add 'in the IFS model' after 'forecast cyclones'.

Done

/l. 330: Remove 'For their part,'.

Done

l. 332: Add 'ly' after 'previous', and a comma after 'CTE'.

Done

l. 333: 'shift' to 'bias' Figure 5: Change 'compared to ERA5 in function of the lead time' to 'relative to ERA5 as a function of lead time'. Replace 'first and third' by 'first to third'. Also, replace 'black lines' by 'black whiskers'.

Done

l. 335: 'when looking at the mean': What is 'mean' referring to here? The ensemble mean? The mean over all lead times considered? Please clarify. If the ensemble mean, the lead time reference is missing.

Mean(s) is referring to the means in the distributions. Correction:"A very weak bias of -0.2 hPa per day is

observed when looking at the means (blue dots)".

l. 336: Add 'error' before 'distribution', and remove 'around the reference' as you can simple say that the MSLP errors are centred around zero.

Done

l. 337: '3 days' à '72 h'

Done

l. 338-339: Is there a specific reason why you are reporting the IQR at 72h lead time? If you want to include in the text, maybe describe how the IQR increases with lead time.

The reason is to give a practical idea to the reader. Correction:"After 72 h of forecast, 50 % of the MSLPEs are between -2.5 hPa and 1.5 hPa and the interquartile range grows linearly of 0.8 hPa until the last lead time of 144 h."

l. 339: Same as above: The question is whether it is reasonable to compare to this deviating range (i.e., frequency distribution) of northern hemispheric cyclone natures.

Answered earlier. Froude does not consider tropical cyclones.

l. 340: 'indicates' to 'indicate'.

Corrected

l. 343: 'event' is rather unspecific. Please clarify what it means. What about using 'cyclone- base time combination' (or 'pair' instead of 'combination')? Admittedly, it sounds more complex, but describes it more precisely.

Correction: "The reliability of the ensemble reforecasts is evaluated for both intensity and location by comparing for each cyclone and at a specific lead time the spread and the mean error of the members".

Figure 6: Suggestion: Replace the first sentence' by 'Spread-skill relationship at 72h lead time with blue shading representing the number of cyclone-base times populating each bin.'. In the last sentence, replace 'event' as in the previous comment, and change 'in each half part' to 'above and below the diagonal, respectively'.

Correction: "Spread-skill relationship at 72h lead time, with blue shading representing the number of cyclones populating each bin. (a) Mean of the TTEs of the ensemble members, denoted $\overline{TTE}$, compared to the spread in location, denoted $\sigma_{loc}$. The bin length is equal to 8 km. (b) Root mean square of the MSLPEs of the ensemble members, denoted by $\langle MSLPE \rangle$, compared to the spread in intensity, denoted $\sigma_{int}$. The bin length is equal to 0.2 hPa. The red curve represents an idealised, perfectly-calibrated set of ensemble reforecasts with a mean error equivalent to the spread. Percentages indicate the proportion of cyclones above and below the diagonal, respectively."

l. 345-346: Remove the two words 'respectively'.

Done

l. 348: Replace the 'in's by brackets around 'Fig. 6x', and replace 'in both cases' by ' for both aspects'.

Done

l. 349: 'trend' to 'relationship',

Changed for 'relationship'.

'reliable' to 'calibrated'.

Reliability is an important aspect of ensemble verification and the spread-error relationship is one way to evaluate it. Perfect reliability can be obtained by statistical post-processing methods (calibration) but here we want to evaluate reliability of the raw ensemble forecast.

l. 350: Since you are making these statements for both plots at once, add about before '60%' as it is '58%' for MSLP. And as above, be more specific and replace 'event'.

Done

l. 353: Suggestion: 'is slightly over-dispersive for most of the cyclone cases' à 'tends to be over-dispersive in most forecasts', and 'remain very poorly predicted' à 'are totally off'.

Done

l. 355: Add 'really' before 'observed', and replace 'the different' by 'other' or 'all'.

Done

l. 357: Add a comma after 'section'.

Done

l. 359-360: Change 'in the spatial distribution, the seasonality' to 'depending on the region, the season'.

Done

l. 362: Replace 'Note that the CDFE having the same unit as the variable considered, the' by 'It should be noted that the CDFE has the same unit as the variable considered. The'

Done

l. 363: Remove two times 'respectively the', and move '(smaller)' after 'greater' in line 362. Figure 7: Move 'at 72 h lead time' after '(b)'.

Done

l. 364: 'On' à 'in'

Done

l. 365: Add 'the shape of' before and 'error' the word 'its'. Also, 'is closer to' à 'better resembles'.

Done

l. 371: 'sections' à 'subsections', remove 'the' and put 'score' in the plural. Also change 'is' to 'are'.

Done

l. 372: Add 'considered' after 'duration'

Done

l. 374: 'spatial distribution' à 'relative frequencies'

We prefer to keep here spatial distribution, as it is the specific aspect considered here.

Figure 8+9+10+11: What does 'from every other categories' mean? Do you test against every other category

separately and then mark the parts where all intersect? Or do you test against all other (i.e., remaining) categories combined? If the latter, replace 'every' by 'all'. In any case, this should be clearly explained here and in the text. Also, replace 'The CDFE score' by 'CDFE scores', and change 'is' to 'are'.

Each category is tested against each other one separately. In this way, two equally populated CDFs are tested each time. Correction: "Differences in the predictability between the regions defined in Fig.8b, for (a) and (c) the total track error (TTE), for (b) and (d) the MSLP error (MSLPE). The statistical significance is tested between each pair of categories. Results appear in thick lines when the category is significantly different from every other. CDFE scores computed from the complete data set are represented by the black circles."

Also, what is plotted when a single lead time has a significant difference (i.e., with the previous and following lead times being insignificant). Is it a circle or a point, or is it possible that nothing is drawn because there is no other point with which a line could be constructed?.

First, the thin curves are plotted. Then the thick (significant) curves are plotted over, considering each set of 2 consecutive instants in a time loop. Consequently, a single instant non-significant would appear as part of a thin curve.

Figure 8: Fig. 8c+d are not needed as they do not provide additional information. Another reason to remove them is that the significant parts change as they are probably now derived testing against only two of other regions, which is a bit confusing (as it is not mentioned in the text). Also, add the black circles in panels (a) and (b).

Done

l. 377: Add 'in terms of cyclone track' after 'predicted'.

Done

l. 378 As for the captions of Fig. 8-11, is it really 'against any other' or 'against all other'?

Against any other.

l. 378-382: Truly an interesting characteristic if it can be clearly attributed to the imprint of the diurnal cycle. However, do you have more supporting evidence for this then 'just' the line plots? For intensity, one can expect such a relationship, but why should it be the same fore location errors? It could be pure coincidence.

Indeed, the diurnal cycle is stronger in intensity, as discussed later.

By construction, the CDFE metric combines the effects of (a) the sharpness of the ensemble (i.e., how distant the errors of the members are spread around the median) and (b) how much the location of the entire error CDF deviates from 0. Can you check the actual CDFs to see if there are differences between 0 and 12 UTC for (a) and (b), respectively?

See Fig.R2-3. The difference is visible for the Middle East in terms of intensity at between 48 h (00 UTC) and 60 h (12 UTC). The result seems to indicate a strong shift of CDF, indicating a real bias rather than a lack of sharpness.

l. 382: 'on' à 'in'

Done

l. 382-385: If Fig. 8c+d are removed, revise sentences and merge with the text describing Fig. 8a+b.

Done

l. 384: 'general' à 'mean'

Done

l.387-388: Significance is either found or not but can't be 'pronounced' from a statistical perspective. You probably mean that it is found for fewer lead times compared to the location results. However, there are also many significant parts for the intensity results, so maybe drop the statement.

Dropped: "Regional differences are observed, in particular between the best and the poorest predicted categories"

l. 383-389: 'The Middle East …': Maybe you can link to Fig. 3a, which shows that there are no very intense cyclones in this region, which is certainly one reason why the intensity is easier to predict. (Maybe it is worth adding relative frequencies for other percentile ranges as panel(s) to Fig. 3, similar to Fig. 3a.)

Correction: "The Middle East is the region in which the intensity of cyclones is the best predicted at each lead time, probably linked with the absence of deep cyclones in this region (see Fig. 3a)."

l. 392: 'firsts' à 'first', and remove 'once again'.

Done

l. 395: 'or' à 'and'

Done

l. 399: change to plural: 'cyclones'.

Done

l. 403: 'the difference remains' à 'differences remain'

Done

l. 406: 'seasons' à 'season', 'The CDFE score' à 'CDFE scores', 'follows' à 'follow'.

Corrected. (every seasons by all seasons).

l. 406: If significance lines are plotted when a category is different from all other categories, how can it be that SON and MAM are almost on top of each other and yet be significantly different between 48h-102h? Do you have an explanation for this?

See Fig.R2-4. The Kolmogorov test is using the maximal (vertical) distance between 2 CDFs. At 72 h lead time for example, the MAM, JJA and SON have similar CDFE but have different CDFs.

Figure 9: Mention here as well why results for SON and MAM are not shown in panel (a)?

Done

l. 408: Different from each other in what sense/variable? MSLPE is shown in Fig. 9b. Are you referring to intensity itself?

Correction: "Secondly, when considering only the winter and summer categories, MSLPE distributions are significantly different at every lead time (not shown). The difference between the two categories increases from 0.1 hPa at 24 h to more than 1 hPa at 144 h lead time."

l. 411 & 429: Keep using 'categories' as before and avoid 'classes' to be consistent.

Done

l. 412: Change 'Fig. 10a' to 'Fig. 10' as it applies to all panels. Add 'Fig. 10a' after 'TTE' in line 413. Figure

10: Remove the word 'pressure'.

Done

l. 414: You make a general statement (about independence) but only show results for a single 12h-period (before peak intensity). Have you checked other periods of the cyclone's lifetime to be so sure?

No, as "the deepening rate" is directly referring to the figure.

l. 419: Here you are comparing to papers that 'showed a poor prediction of the intensity', but Fig. 10d presents deepening rates. I suggest to move this sentence to where Fig. 10b is discussed.

Done

l. 421: 'day 4.5' à '108 h'. Also, where is it shown in the paper that 'in our data set is slightly over-predicted from day 4.5 onward'? Are you referring to Fig. 5b? However, the plots exhibit an over-prediction on average even before 108h.

Correction: "the intensity of deep storms in our data set is slightly over-predicted from 108 h onward (not shown), while it is slightly under-predicted in these two previous studies". You cannot know if the values describe an under or an over-prediction from the figure, as the CDFE is making the results absolutes.

l. 421-424: Maybe is it not just a matter of sampling cases but also of the different regions that are considered.

Added

l. 425: 'from' à 'at'

Done

l. 427: 'the previous part' à 'Section 5.2'. Figure 11: Add 'moving cyclones, respectively' after 'fastest'.

Done

l. 430: 'based on' à 'in'

Done

l. 431: 'spatial distributions' à 'patterns of relative frequency'

For the same reasons as earlier, we prefer to keep spatial distributions here

l. 432: Remove 'examined in Section 3.4' as the categorisation here is applied to a different (namely the reforecast) data set. When removed, also change 'for the' à 'for a'.

Done

l. 434: 'from the beginning of the forecast' is only true for the fast category.

Changed to "are statistically significant from 6 to 66 h lead time"

l. 445: 'constant' à 'fixed'

Done

l. 447: Inconsistent years when compared to line 95.

Corrected

l. 450: 'Comparatively': Do you mean 'In comparison' or 'in contrast'? If the former, the rest of the sentence needs a comparative form (e.g. 'higher').

Correction: "In comparison to previous studies, a higher density".

l. 456: 'good reliability of the' à 'well calibrated'

Reliability have a strong meaning here.

l. 457: Add 'on average' after 'observed'.

Done

l. 458: Depending on whether you have followed the suggestion above, change 'for most of the cases' to 'in most forecasts'. Also, add 'median and mean' before 'errors'.

Done

l. 460: In my view, the newly developed CDFE score is not a generalization of the CRPS but a different application of the same concept, namely applied to forecast errors instead of actual forecast values.

Changed to "by introducing a newly-defined CDFE score, which is the CRPS applied to the error distribution."

Also, remove 'probabilistic score' as it is part of 'CRPS' already, and replace 'case' by 'cyclone forecasts'.

Done

l. 462-464: Here and in lines 310-311: Why did you split the error growth rate at these lead times? Is there a particular reason?

The precise values were changed to "3 days" and "longer lead times".

l. 462: Add 'nearly' or 'almost' before 'constant'.

Done

l. 464: Add 'with lead time' after 'errors', and 'progressive' before 'saturation'.

Done

l. 470: 'the key' à 'a key'

Done

l. 471: Remove second comma.

Done

l. 472: 'important by causing' à 'considerable, as they can cause'.

Done

l. 480: As before, your study does not distinguish between different stages of cyclone life time, nor cyclogenesis nor cyclolysis. Therefore, replace 'cyclogeneses' by 'cyclones'.

Done

l. 484: Add a comma after 'e.g.'.

Done

l. 488: Very minor, but why do you reverse the order of the categorisations applied in your study?

The order is set from the most to the least impacting.

l. 491: 'firsts' à 'first'

Done

l. 491-492: Move 'at high lead times (several days)' to after 'cyclones', and add 'part of' after 'responsible to'.

Correction: "Indeed, both the representation of latent heat release in the first forecast hours, and the location of Rossby wave breaking at high lead times (several days), may be responsible to part of the loss of predictability of Mediterranean cyclones." " at high lead times" cannot be moved here.

l. 500: Add '.'

Done

l. 501: 'Is' à 'It'

Done

**Additional figures**

[Figure]

Figure R2-1: Median speeds of Mediterranean cyclones. Median is in red and mean in black.
.

[Figure]

Figure R2-2: Average directions of the cyclone tracks (whole dataset)
.

[Figure]

Figure R2-3: CDFs of errors in terms of TTE (left) and MSLPE (right) for 0 h lead time (top) and 12 h lead time (bottom) for different region-based categories.

.

[Figure]

Figure R2-4: CDFs of errors in terms of MSLPE at 72 h lead time for different season-based categories.

.

[Figure]

Figure R2-5: Comparison between Ayrault and VDG tracks obtained from ERA5: distance (a and c) and absolute MSLP difference (b and d). (a) and (b) show the differences in the original data sets, while (c) and (d) show the differences after removing tracks with a distance reaching more than 40 km (final data sets). 0 h lead time corresponds to the time of minimum MSLP in VDG.

.

---

## Referee Report (RR1)

**Review of '*What determines the predictability of a Mediterranean cyclone?*' by Benjamin Doiteau, Florian Pantillon, Matthieu Plu, Laurent Descamps, and Thomas Rieutord**

In the revised version of their manuscript, the authors have made great efforts to take into account the general comments made in the first review. They rephrased the title to more accurately reflect the specific aspects addressed in their predictability study. As for the comments on storm nature, they emphasize that their focus is on the extratropical storms and argue that medicanes would only contribute with a negligible fraction. I would like to clarify that my comment was rather meant not to simply distinguish into extratropical storms and medicanes, but to analyse a bit more how the full spectrum looks like, i.e., including the hybrid stages. It would have been interesting to also condition on the storm nature, because regional differences may be related to it. Even though the authors have refrained from taking up this aspect, I think that at least the clarification made throughout the paper that the majority of the storms are 'extratropical' is helpful for the reader.

The main methodological concern, raised by both reviewers, namely the inconsistency between tracking algorithms, has been seriously addressed. The authors answered many questions related to it and clarified it in the manuscript as well. An extra analysis has been done, applying the VDG algorithm to the ERA5 dataset, resulting in the additional section 2.5, which presents the results of the comparison of the two tracking techniques.

Since the revision has improved the quality of the paper in a good way, and the authors have replied to the review comments in a satisfactory manner, I now recommend the paper for publication after consideration of the minor specific comments below.

*For future revisions, when you prepare a response to a reviewer, please give line number(s) of the text you cite from your revised manuscript in the response to reviewers. Otherwise, one has do the search for the position in the revised document oneself. Thanks!*

**Specific comments**

l. 104: Add "(125-km horizontal resolution)" after "model data".

l. 143: Please add again the word "reforecasts" after "ensemble".

l. 163: Please check my comment to line 158 in the previous version of your manuscript. In the tracked changes document, I can see that you (probably accidentally) removed the sentence you indicated as corrected in your reply to my comments.

l. 188: Replace "non-identical" by "different".

l. 189-190: To make the reader aware of it, please add a sentence on the fact that your comparison on the ERA5 dataset between VDG and AYRAULT has a limitation in terms of storm occurrence. VDG can only identify occurrence if it was found by AYRAULT before.

l. 312-313: I'm not convinced by the statements you make in response to my comment (l. 296-297 in the original version of the manuscript). I don't see why it would be "very unlikely" that extratropical cyclones enter mutliple of the regions you defined throughout their lifetime. If they are embedded in Rossby waves, they can easily get steered over long distances. And, again, I recommend not using the phrase "suggesting two different processes of cyclogenesis" as you are not distinguishing between different stages of a cyclone's lifetime in your study. Keep it on the "occurrence"-level, instead of speculating about stages!

l. 327: Replace "increases of" by "increases by".

l. 329-332: I know what you want to say, but I still find it hard to read and digest this sentence. Consider checking and simplyfing it further to help the reader. To me, the explanations in brackets (i.e., …) are more confusing then helping as they address a totally different question (namely track divergence, while the main text discusses cyclone numbers changing with lead times).

l. 368: On your response to my comment on l. 349 in the previous version of the manuscript: The word "calibration" is often degraded to a statistical post-processing method (i.e., a methodolocial term), but it is actually much more than that. In statistics, it means a joint property of the predictions and the events that materialize, and is thus equivalent to the word "reliable".

l. 452: Given that CDFEs yields absolute values, using 'over-predicted' seems not appropriate then. Please reword.

---

## Author Response (AR2)

**Response to Reviewer #2, 2nd Round**

We thank the reviewer for his/her time and constructive suggestions. We comply with the proposed changes. In the following, the comments made by the reviewer appear in black, while our replies are in blue and the changes in the text are in quotation marks.

**General:**

In the revised version of their manuscript, the authors have made great efforts to take into account the general comments made in the first review.

They rephrased the title to more accurately reflect the specific aspects addressed in their predictability study. As for the comments on storm nature, they emphasize that their focus is on the extratropical storms and argue that medicanes would only contribute with a negligible fraction. I would like to clarify that my comment was rather meant not to simply distinguish into extratropical storms and medicanes, but to analyse a bit more how the full spectrum looks like, *i.e.*, including the hybrid stages. It would have been interesting to also condition on the storm nature, because regional differences may be related to it. Even though the authors have refrained from taking up this aspect, I think that at least the clarification made throughout the paper that the majority of the storms are 'extratropical' is helpful for the reader.

The main methodological concern, raised by both reviewers, namely the inconsistency between tracking algorithms, has been seriously addressed. The authors answered many questions related to it and clarified it in the manuscript as well. An extra analysis has been done, applying the VDG algorithm to the ERA5 dataset, resulting in the additional section 2.5, which presents the results of the comparison of the two tracking techniques.

Since the revision has improved the quality of the paper in a good way, and the authors have replied to the review comments in a satisfactory manner, I now recommend the paper for publication after consideration of the minor specific comments below.

*For future revisions, when you prepare a response to a reviewer, please give line number(s) of the text you cite from your revised manuscript in the response to reviewers. Otherwise, one has do the search for the position in the revised document oneself. Thanks!*

**Specific comments:**

l. 104: Add "(125-km horizontal resolution)" after "model data".

l. 110 of the track changes document: Done

l. 143: Please add again the word "reforecasts" after "ensemble".

l. 148 of the track changes document: Changed for "Data for the predicted tracks: the IFS ensemble reforecasts", to avoid the repetition. 2.4. title also changed for consistency: "Tracking method for the predicted tracks: the VDG algorithm"

l. 163: Please check my comment to line 158 in the previous version of your manuscript: "*l. 158: Replace 'track predicted cyclones' by 'tracks predicted for cyclones'.*" In the tracked changes document, I can see that you (probably accidentally) removed the sentence you indicated as corrected in your reply to my comments.

l. 168 of the track changes document: We find that the following sentence: "This characteristic is particularly useful when it comes to tracks predicted for cyclones that were previously identified in the reference data set." is not easily understandable. Changed to: "This characteristic is particularly useful when it comes to detect cyclones in the reforecasts from the location of the reference tracks."

l. 188: Replace "non-identical" by "different".

l. 195 of the track changes document: Done

l. 189-190: To make the reader aware of it, please add a sentence on the fact that your comparison on the ERA5 dataset between VDG and AYRAULT has a limitation in terms of storm occurrence. VDG can only identify occurrence if it was found by AYRAULT before.

The purpose of this paragraph is not to compare two independent tracking techniques, but to see if the tracks detected in VDG and AYRAULT are indeed the same cyclones. Therefore, we do not think that repeating the limitations of VDG is relevant in this part.

l. 312-313: I'm not convinced by the statements you make in response to my comment (l. 296-297 in the original version of the manuscript). I don't see why it would be "very unlikely" that extratropical cyclones enter multiple of the regions you defined throughout their lifetime. If they are embedded in Rossby waves, they can easily get steered over long distances. And, again, I recommend not using the phrase "suggesting two different processes of cyclogenesis" as you are not distinguishing between different stages of a cyclone's lifetime in your study. Keep it on the "occurrence"-level, instead of speculating about stages!

l. 322 of the track changes document: "Cyclogenesis" changed to "cyclones". The aforementioned comment and the answer we made (italic) are the following:

> l. 296-297: As you do not distinguish between different stages of a cyclone lifetime in your study, I would refrain from linking different motion speed-categories to differences in processes of cyclogenesis. It may be that a cyclone forms in one part of a region but then moves into another part of the same region, while changing motion speed. Therefore, here and throughout the paper, I recommend to not speculate on cyclogenesis, if it is not specifically analysed.
> *Indeed, the motion speed of a cyclone may vary during its lifetime. However, the areas highlighted on Figure 4a are clearly distinct from each other and it is very unlikely the same cyclone would move from one to the other. Also, note that we do not explicitly refer to the cyclogenesis area but more broadly to where cyclones arrive from (the Sahara, the Atlantic).*

We identify three arguable points:

1. The possibility for a cyclone to move from one region to another ("It may be that a cyclone forms in one part of a region but then moves into another part of the same region, while changing motion speed." in the previous comment and "I'm not convinced by the statements you make in response to my comment" in the current comment)

2. The Rossby waves steering storms over long distance ("If they are embedded in Rossby waves, they can easily get steered over long distances" in the current comment)

3. The speculative claims made in the manuscript about the processes ("I recommend not using the phrase *suggesting two different processes of cyclogenesis*", in the current comment)

For which will we attempt a better answer:

1. Our claim that "it is very unlikely the same cyclone would move from one to the other" is actually based on our experience with the AYRAULT algorithm: most of the tracks are short in space and time, and we could find just a few examples of a cyclone crossing several regions in our dataset.

2. It is true, but not for every region. The Mediterranean Sea is embedded by mountain ranges, therefore the cyclones cannot be always steered easily over long distances without discontinuity.

3. Despite our current study cannot address the link between processes and predictability, we believe it will be a very interesting study to do in the future, and we would like to suggest potential directions for this research, based on our experience. This was our intention when we used the word "suggesting". However to avoid any confusion, "different types of cyclogenesis" is changed to "different types of cyclones".

l. 327: Replace "increases of" by "increases by".

l. 336 of the track changes document: Done

l. 329-332: I know what you want to say, but I still find it hard to read and digest this sentence. Consider checking and simplifying it further to help the reader. To me, the explanations in brackets (i.e., . . . ) are more confusing then helping as they address a totally different question (namely track divergence, while the main text discusses cyclone numbers changing with lead times).

The track divergence is essential, a strong (weak) track divergence would imply great (small) errors. As the track divergence increases almost always with lead time, the link between proportion of cyclones followed since early/long lead time is crucial. The whole paragraph is changed to: "The distribution of TTEs is presented for each lead time up to 144 h (Fig. ??a). Both median error and interquartile range increase as lead time increases. For instance, after 72 h lead time, 50 % of the TTEs spans from 80 km to 220 km. Interestingly, the error growth is slower than linear and seems to exhibit two phases: during the first 78 h, the median TTE increases by about 40 km per day, while it increases at a smaller rate of about 20 km per day from 84 h lead time onward. This behaviour can be explained by two different reasons. Firstly, the construction of VDG constrains the tracking to start near the reference track. Given that the median lifetime of the cyclones of our dataset is 42 h, as lead time increases, the proportion of cyclones tracked from early lead times (where the forecast track may have diverged from the reference track) decreases, compared to those tracked from longer lead times (where the forecast track remains close to the reference track). As a result, the error growth tends to slow down as lead time increases. Second, the phenomenon of error saturation also plays a role. For long enough lead times, an ensemble forecast is expected to converge to the climatological distribution. Consequently, the mean and median errors are anticipated to increase at a slower rate at long lead times and saturate ultimately at constant values."

l. 368: On your response to my comment on l. 349 in the previous version of the manuscript: the word "calibration" is often degraded to a statistical post-processing method (i.e., a methodological term), but it is actually much more than that. In statistics, it means a joint property of the predictions and the events that materialize, and is thus equivalent to the word "reliable".

The aforementioned comment and the answer we made (italic) are the following:

> 'reliable' to 'calibrated'.
> *Reliability is an important aspect of ensemble verification, and the spread-error relationship is one way to evaluate it. Perfect reliability can be obtained by statistical post-processing methods (calibration) but here we want to evaluate reliability of the raw ensemble forecast.*

If the two words are equivalent, we prefer to stay on the word "reliable" for consistency.

l. 452: Given that CDFEs yields absolute values, using 'over-predicted' seems not appropriate then. Please reword.

l. 463 of the track changes document: The word over-predicted mentioned here does not refer to the CDFE metric, but to the errors of MSLP (not shown). Changed to: "However, it should be noted that on average, the forecast intensity of deep storms in our dataset is slightly too strong from 108 h onward (not shown), while it is slightly too weak in these two previous studies."